# Prediction-aware Learning in Multi-agent Systems

**Aymeric Capitaine** [1]  **Etienne Boursier** [2]  **Eric Moulines** [1]  **Michael I. Jordan** [3]  **Alain Durmus** [1]

## Abstract

The framework of uncoupled online learning in multiplayer games has made significant progress in recent years. In particular, the development of *time-varying games* has considerably expanded its modeling capabilities. However, current regret bounds quickly become vacuous when the game undergoes significant variations over time, even when these variations are easy to predict. Intuitively, the ability of players to forecast future payoffs should lead to tighter guarantees, yet existing approaches fail to incorporate this aspect. This work aims to fill this gap by introducing a novel *prediction-aware* framework for time-varying games, where agents can forecast future payoffs and adapt their strategies accordingly. In this framework, payoffs depend on an underlying state of nature that agents predict in an online manner. To leverage these predictions, we propose the `POMWU` algorithm, a contextual extension of the optimistic Multiplicative Weight Update algorithm, for which we establish theoretical guarantees on social welfare and convergence to equilibrium. Our results demonstrate that, under bounded prediction errors, the proposed framework achieves performance comparable to the static setting. Finally, we empirically demonstrate the effectiveness of `POMWU` in a traffic routing experiment.

## 1. Introduction.

The framework of uncoupled online learning in multiplayer games has sparked a lot of interest for its ability to realistically model the interactions of rational players engaged in a dynamic game. Since the seminal works of Foster and Vohra (1997); Freund and Schapire (1999); Hart and Mas-Colell (2000a), progress has been made towards obtaining fast convergence rates for different equilibrium concepts, including coarse correlated equilibrium (Syrgkanis et al., 2015; Foster et al., 2016; Daskalakis et al., 2021; Piliouras et al., 2022; Farina et al., 2022) correlated equilibrium (Chen and Peng, 2020; Anagnostides et al., 2022a;b; Peng and Rubinstein, 2023) and Nash equilibrium (Anagnostides et al., 2022c). However, most of these works assume that the game remains constant over time.

Only recent studies have begun to consider time-varying games, in both two-player zero-sum games (Zhang et al., 2022) and multiplayer general-sum games (Duvocelle et al., 2018; Anagnostides et al., 2024). Following methods initially developed by the online optimization community (Chiang et al., 2012; Rakhlin and Sridharan, 2013), these studies bound the dynamic regret incurred by players with measures of the *time-variation* of the underlying game. While this approach looks satisfactory at first glance, it is not hard to come up with simple examples for which the variation is important–making the above mentioned bounds vacuous–yet very simple to predict. In Example 1, we exhibit a simple instance of time-varying game where the regret bounds derived in Zhang et al. (2022) grow linearly with the horizon $T > 0$. However, the dynamic underlying the payoff matrices is entirely deterministic, and knowing it would result in a constant regret. This highlights that the current time-varying framework fails to account for any *predictive capacity* of the agents. This is all the more surprising as predictive models become ubiquitous in numerous economic sectors (Jordan and Mitchell, 2015; Gogas and Papadimitriou, 2021; Hinton and Jordan, 2024), making it likely for strategic agents to possess a forecasting ability regarding their future payoffs. This work aims to fill the gap, by asking the following question:

> How does the quality of predictions made by rational agents in time-varying games regarding their future payoffs affects social welfare, as well as the convergence to equilibrium ?

**Contributions.**  We address this question with the following contributions.

[1]Centre de Mathématiques Appliquées – CNRS – École polytechnique – Palaiseau, 91120, France [2]Inria Saclay, Université Paris Saclay, LMO - Orsay, 91400, France [3]Inria Paris, Ecole Normale Supérieure, PSL Research University - Paris, 75, France. Correspondence to: Aymeric Capitaine <firstname.lastname@polytechnique.edu>.

*Proceedings of the 42$^{nd}$ International Conference on Machine Learning*, Vancouver, Canada. PMLR 267, 2025. Copyright 2025 by the author(s).

- First, we introduce the new *prediction-aware* learning framework, where players forecast future payoffs in an online fashion and design their strategies accordingly. In a nutshell, we build on the contextual setting proposed by Sessa et al. (2021) by introducing an underlying state of nature, either adversarially or stochastically drawn, which determines the payoff of all agents. They play a time-varying game which can be decomposed into three stages. First, each player forecasts the current state of nature based on their local predictor before picking an action in the game. Then, they observe their payoff and the actual state of nature. Finally, they update their policy and predictor based on these new observations. Augmenting uncoupled learning in games with contexts and predictions requires to introduce new regret and equilibrium concepts. In particular, we extend correlated equilibrium (Aumann, 1987) to our framework.

- Second, we propose an algorithm called POMWU—which a contextual extension of the optimistic Multiplicative Weight Update algorithm (Daskalakis et al., 2021)—allowing players to leverage their prediction about the state of nature. In particular, we show that if all players use POMWU, we match the results of Syrgkanis et al. (2015) established for static games, regarding social welfare (Corollary 3), equilibrium convergence (Corollary 2) and robustness in the adversarial setting (Proposition 8) up to a factor that depends polynomially on the number of prediction errors by players. Thus, when predictions errors are bounded by a constant (which is the case under realizability, see Daniely et al., 2014), our bounds match the guarantees on social welfare and equilibrium convergence for static games. Our analysis builds upon a new notion of *contextual* Regret bounded by Variation Utility (RVU) which bounds contextual regret by the sum of the length of the context-specific sequences of feedbacks and strategies. Indeed, a naive application of the standard RVU framework results in looser bounds.

**Additional related works.** The problem tackled in this work relates with several lines of research in game theory and online optimization. On the one hand, the contextual optimization literature (Donti et al., 2019; Elmachtoub and Grigas, 2020; Bennouna et al., 2024) has considered the problem of minimizing an objective function defined by an unobserved random context, which the optimizer can predict via a regression function. This idea has also been studied in the contextual bandit framework (Lattimore and Szepesvári, 2020) with noisy contexts (Kirschner and Krause, 2019; Yang and Ren, 2021; Nelson et al., 2022; Guo et al., 2024). However, none of these works consider the multi-agent setting, where the optimizer interacts with other agents during the learning process. On the other hand, recent studies in

game theory have incorporated the idea of an underlying state of nature jointly determining the payoffs of players. While Sessa et al. (2021); Maddux and Kamgarpour (2024) studies the contextual version of uncoupled learning in multiplayer games, Lauffer et al. (2023); Harris et al. (2024) focuses on Stackelberg games with side information. However, these works assume that the context is revealed to players at the beginning of each period, unlike ours where players have to predict the context before moving. In the end, the *social learning* framework might be the one that relates the most to ours. Pioneered by the work of Banerjee (1992); Bikhchandani et al. (1992); Smith and Sørensen (2000), it features agents receiving private signals about a true, unobserved state of nature. These agents are able to learn from both their signal and the actions played by other players, which reflect their signals Chamley (2004). Most of the social learning literature has been devoted to analyzing the resulting collective behaviors, such as cascading and herding phenomena (Mossel et al., 2020). While recent studies have broadened the analytical toolbox of social learning by considering for instance time-varying states of nature (Frongillo et al., 2011; Boursier et al., 2022; Levy et al., 2024), it mostly relies on very strong assumptions (e.g. a binary state and binary actions, Mossel et al., 2020) and a Bayesian modeling where all agents share a common prior about the state of nature's distribution. In contrast, we believe that the uncoupled learning framework (Hart and Mas-Colell, 2000b; 2003; Daskalakis et al., 2011) upon which our work relies is a more general setting for studying this question, and allows to study more natural equilibrium concepts such as correlated equilibria (Aumann, 1987) with stronger guarantees.

**Organization.** This work is organized as follows. In Section 2, we present our model, notion of regret and main assumptions. In Section 3, we introduce the POMWU algorithm and establish the convergence of social welfare and individual utilities. In Section 4, we empirically demonstrate the performance of POMWU on the Sioux Falls routing problem (LeBlanc et al., 1975).

**Example 1.** *Consider the two-players setting in (Zhang et al., 2022) where $\mathcal{X} \in \mathbb{R}^n$ and $\mathcal{Y} \in \mathbb{R}^m$ are respectively the strategy spaces of player $x$ and $y$, $A_t \in [-1, 1]^{n \times m}$ is their time-varying payoff matrix and $\mathcal{E}_t \subset \mathcal{X} \times \mathcal{Y}$ is the set of Nash equilibria at time $t \in [T]$. The two measures of variations considered in (Zhang et al. 2022, and up to minor modifications Anagnostides et al. 2024) are*

$$P_T = \min_{\mathcal{E}_1 \times \ldots \times \mathcal{E}_T} \sum_{t \in [T]} \left( \left\| x_t^\star - x_{t-1}^\star \right\|_1 + \left\| y_t^\star - y_{t-1}^\star \right\|_1 \right),$$

*and*

$$V_T = \sum_{t \in [T]} \left\| A_t - A_{t-1} \right\|_\infty^2,$$

*which are respectively the variation of Nash equilibria and the variation of payoff matrices.* [Zhang et al. (2022, Theorem 6)](#) *show that the dynamic regret can be bounded by*

$$\widetilde{\mathcal{O}}(\min(\sqrt{(1 + P_T)(1 + V_T)} + P_T, 1 + W_T)) , \quad (1)$$

*where* $W_T = \sum_{t \in [T]} \left\| A_t - T^{-1} \sum_{\tau \in [T]} A_\tau \right\|_\infty = \Omega(V_T)$. *On the other hand, if we consider for any* $t \in [T]$, $A_t = B + (-1)^t C$ *where*

$$B = \frac{1}{2}\begin{pmatrix} 1 & 1 \\ 1 & 1 \end{pmatrix}, \ C = \frac{1}{2}\begin{pmatrix} 1 & 1 \\ -1 & -1 \end{pmatrix},$$

*it is not hard to check that*

$$\mathcal{E}_t = \begin{cases} \{(1, 0), (\frac{1}{2}, \frac{1}{2})\} & \text{if } t \text{ is even} \\ \{(0, 1), (\frac{1}{2}, \frac{1}{2})\} & \text{otherwise} \end{cases}.$$

*This implies that* $P_T = 2T$. *Likewise, one can verify that* $V_T = T$, *so the bound in* (1) *grows linearly with* $T$. *At the same time, we remark that* $Y_t = -Y_{t-1}$ *with* $Y_t = A_t - A_{t-1}$. *This shows that* $(A_t)_{t \in [T]}$ *is a deterministic process (more precisely, a deterministic ARIMA(1,1,0) process (*[Hamilton, 2020](#)*)).*

## 2. Model.

**Notation.** In what follows, we denote the $\ell$-th coordinate of any vector $x \in \mathbb{R}^d$ by $x[\ell] \in \mathbb{R}$. Likewise, the $\ell$-th row of any matrix $\mathbf{X} \in \mathbb{R}^{d \times K}$ is denoted by $\mathbf{X}[\ell] \in \mathbb{R}^K$. For any vectors $(x, y) \in \mathbb{R}^d \times \mathbb{R}^d$, we write $\langle x, y \rangle = x^\top y$ the standard euclidian inner product and $x.y = (x[1] \, y[1], \ldots, x[d] \, y[d])^\top$ the Hadamard product. $\mathscr{P}(\mathcal{A})$ denotes the set of probability measures over a measurable space $\mathcal{A}$, and $\Delta_K = \{w \in \mathbb{R}^K : \forall \ell \in [K], \ w^j[\ell] \geqslant 0 \text{ and } \sum_{\ell=1}^K w^j[\ell] = 1\}$ the simplex of dimension $K > 0$. When $\mathcal{A} = \mathcal{A}^1 \times \ldots \times \mathcal{A}^J$ is the product of $J > 0$ spaces, we write $\mathcal{A}^{-j} = \mathcal{A}^1 \times \ldots \times \mathcal{A}^{j-1} \times \mathcal{A}^{j+1} \times \ldots \mathcal{A}^J$ for any $j \in [J]$, so $\mathcal{A} = \mathcal{A}^j \times \mathcal{A}^{-j}$. For any $\mathbf{w} \in \mathscr{P}(\mathcal{A})$, we write $\mathbb{E}_{a \sim \mathbf{w}}[a] = \int a \, d\mathbf{w}(a)$ the associated expectation. When the context is clear, we rather write $\mathbb{E}_{\mathbf{w}}$ instead of $\mathbb{E}_{\mathbf{a} \sim \mathbf{w}}$. When $\mathbf{w} = w^1 \otimes \ldots \otimes w^J$ is a product of $J > 0$ measures, we define for any $j \in [J]$ $\mathbf{w}^{-j} = w^1 \otimes \ldots w^{j-1} \otimes w^{j+1} \otimes \ldots \otimes w^J$ and $\mathbb{E}_{\mathbf{w}^{-j}}$ the associated expectation operator.

**Setting.** We consider a set of $J > 0$ agents denoted by $[J]$. We suppose that each agent has access to an action set $\mathcal{A}^j = \{a_1^j, \ldots, a_K^j\}$ with $|\mathcal{A}^j| = K$. In addition, we assume that the cost function of agent $j \in [J]$ is given for $Z \in \mathcal{Z} \subseteq \mathbb{R}^d$ and $\phi^j : \mathcal{A} \to \mathbb{R}^d$ by:

$$c^j(\mathbf{w}, Z) = \mathbb{E}_{\mathbf{a} \sim \mathbf{w}}\left[ \langle \phi^j(\mathbf{a}), Z \rangle \right] , \quad (2)$$

where $\mathbf{w} \in \mathscr{P}(\mathcal{A})$. Typically, we will consider $\mathbf{w} = w^1 \otimes \ldots \otimes w^J$ where $w^j \in \Delta_K$ is a mixed strategy played by

$j \in [J]$. This cost function is flexible and is customary in contextual optimization ([Sadana et al., 2024](#)) and contextual bandit ([Li et al., 2010](#); [Lattimore and Szepesvári, 2020](#)). (2), $\phi^j$ represents a standard payoff function, while $Z \in \mathcal{Z}$ can be interpreted as a state of nature that linearly influences preferences. Note that a time-varying game can easily be constructed by considering a sequence of states of nature $(Z_1, \ldots, Z_T) \in \mathcal{Z}^T$ for $T > 0$. We rewrite (2) in a more compact way with the following lemma.

**Lemma 1.** *Let* $j \in [J]$, $\mathbf{w} \in \mathscr{P}(\mathcal{A})$ *with* $\mathbf{w} = w^j \otimes \mathbf{w}^{-j}$ *and* $\Phi^j(\mathbf{w}^{-j}) = (\mathbb{E}_{\mathbf{w}^{-j}}[\phi^j(a_k^j, \mathbf{a}^{-j})[\ell]])_{\ell,k} \in \mathbb{R}^{d \times K}$. *We have:*

$$c^j(\mathbf{w}, Z) = \left\langle Z, \Phi^j(\mathbf{w}^{-j}) w^j \right\rangle .$$

In Lemma [1](#), $\Phi^j(\mathbf{w}^{-j})$ is a matrix whose column $k$ contains the cost of playing the pure action $a_k^j$ when opponents play their mixed-strategy $\mathbf{w}^{-j}$. This quantity appears naturally in online learning for games ([Syrgkanis et al., 2015](#)). Note that by Lemma [1](#), $\Phi^j(\mathbf{w}^{-j})$ entirely determines $c^j$, so having access to this matrix is equivalent to having access to $c^j$. Moreover, Lemma [1](#) stresses that $c^j$ is conveniently linear in $w^j \in \Delta_K$ for any $j \in [J]$.

We introduce the two following assumptions for the rest of the analysis.

**H1.** *For any* $j \in [J]$, $\mathbf{a} \in \mathcal{A}$ *and* $Z \in \mathcal{Z}$, $\left| \langle Z, \phi^j(\mathbf{a}) \rangle \right| \leqslant 1$.

This boundedness assumption is usual in learning in games and more generally in online learning ([Hazan et al., 2014](#)). In particular, **H**[1](#) ensures that for any $j \in [J]$, $\mathbf{w} \in \mathscr{P}(\mathcal{A})$ and $Z \in \mathcal{Z}$, $c^j(\mathbf{w}, Z) \leqslant 1$.

**H2.** *The set* $\mathcal{Z}$ *is finite:* $\mathcal{Z} = \{z_1, \ldots, z_m\}$ *for* $m > 0$.

Assuming a finite context set is customary in bandit theory ([Lattimore and Szepesvári, 2020](#); [Slivkins et al., 2019](#)) and game theory ([Kamenica and Gentzkow, 2011](#); [Kamenica, 2019](#)), and is often relevant in practical settings. We believe that the analysis to an infinite context set is possible, but such an extension falls outside the scope of the current paper, as it would require introducing fundamentally different concepts and proof techniques, see Appendix [B](#) for further discussion.

We assume that agents play a time-varying game, which is determined by a sequence of states of nature $(Z_1, \ldots, Z_T) \in \mathcal{Z}^T$ of length $T > 0$. At the beginning of each period $t \in [T]$, nature draws a state of nature $Z_t \in \mathcal{Z}$, which is not revealed to agents, while each player $j \in [J]$ receives a signal $\hat{Z}_t^j \in \mathcal{Z}$ about this state. They then select a strategy $w_t^j \in \Delta_K$ based on this signal. Finally, each agent $j$ get as a feedback the cost matrix $\Phi^j(\mathbf{w}_t^{-j})$ as well as the actual state of nature $Z_t$.

**Remark 1.** *In many practical settings, the private signals* $\hat{Z}_t^j \in \mathcal{Z}$ *for* $j \in [J]$ *and* $t \in [T]$ *are predictions made by*

*supervised learning algorithms. In this case, at the beginning of each round $t \in [T]$, each agent $j \in [J]$ observes covariates $X_t^j \in \mathcal{X}$ and makes a prediction*

$$\hat{Z}_t^j = g_t^j(X_t^j) \,,$$

*where $g_t^j \in \mathcal{G} \subset \{g : \mathcal{X} \to \mathcal{Z}\}$ is some prediction algorithm based on the history of observations up to time $t$. Under H2, this situation corresponds to multiclass online learning, for which several theoretical results are available in the litterature ([Daniely et al., 2014](); [Daniely and Shalev-Shwartz, 2014]()).*

To formally describe the game, we define $\Pi^j$ as the set of policies $\pi^j : (\cup_{t \in [T]} \mathcal{H}_t^j) \times \mathcal{Z} \to \Delta_K$ for player $j \in [J]$, where $\mathcal{H}_t^j$ is the set of histories at time $t \in [T]$ with elements $h_\tau^j = \{\Phi^j(\mathbf{w}_\tau^{-j}), Z_t\}_{1 \leqslant \tau \leqslant t}$. At the beginning of the game, $h_0^j = \emptyset$. Then for any $t \in [T]$,

1. Each agent $j \in [J]$ observes a private signal $\hat{Z}_t^j \in \mathcal{Z}$, and picks a mixed strategy $w_t^j \in \Delta_K$ where $w_t^j$ is the output of a policy $\pi_t^j = \pi^j(h_{t-1}^j, \cdot) : \mathcal{Z} \to \Delta_K$, that is $w_t^j = \pi_t^j(\hat{Z}_t^j)$.

2. Each agent $j$ incurs a cost $\langle Z_t, \Phi^j(\mathbf{w}_t^{-j})w_t^j \rangle$, and gets as a feedback $(Z_t, \Phi^j(\mathbf{w}_t^{-j}))$. They then update $h_t = h_{t-1} \cup \{\Phi^j(\mathbf{w}_t^{-j}), Z_t\}$.

**Remark 1** ([continuing]() from p. [3]()). *In the case where private signals are predictions from an online algorithm, agents train policies $\kappa^j : \mathcal{X} \to \Delta_K$ mapping covariates to strategies. Indeed, for any $j \in [J]$ and $t \in [T]$:*

$$w_t^j = \pi_t^j(\hat{Z}_t^j) = (\pi_t^j \circ g_t^j)(X_t^j) = \kappa_t^j(X_t^j) \,.$$

We consider the standard full-information feedback setting, where each player $j \in [J]$ observes $\Phi^j(\mathbf{w}_t^j)$. We believe that extending our results to bandit feedback – i.e., when agents only observe the reward from their realized action– ([Foster et al., 2016]()) is feasible, though it would require additional technical refinements.

**Regrets.** We now present the two regret concepts used in this paper to quantify the optimality of a policy $\pi^j \in \Pi^j$. They are essentially contextual versions of the classic external ([Zinkevich](), [2003]()) and swap ([Blum and Mansour](), [2007]()) regrets. In what follows, $\mathcal{T}^z = \{t \in [T] : Z_t = z\}$ denotes the timesteps at which $z \in \mathcal{Z}$.

First, following [Sessa et al. (2021)](), we introduce a contextual external regret. For any $j \in [J]$, given a fixed sequence of opponent strategies $(\mathbf{w}_t^{-j})_{t \in [T]}$, we denote by $\pi_\star^j : \mathcal{Z} \to \Delta_K$ the static comparator which satisfies

$$\sum_{t \in \mathcal{T}^z} c^j(\pi_\star^j(z), \mathbf{w}_t^{-j}, Z_t) \leqslant \sum_{t \in \mathcal{T}^z} c^j(w, \mathbf{w}_t^{-j}, Z_t) \,, \quad (3)$$

for any $z \in \mathcal{Z}$ and $w \in \mathcal{P}(\mathcal{A}^j)$. The comparator $\pi_\star^j$ maps each context $z$ to the best action in hindsight on the time steps when $z$ was observed. Denoting $w_t^j = \pi_t^j(\hat{Z}_t^j)$ the strategy of agent $j$ for any $t \in [T]$, we then define the following contextual external regret:

$$\mathfrak{R}_T^j = \sum_{t \in [T]} \left[ c^j(w_t^j, \mathbf{w}_t^{-j}, Z_t) - c^j(\pi_\star^j(Z_t), \mathbf{w}_t^{-j}, Z_t) \right] \,. \tag{4}$$

Note that $\mathfrak{R}_T^j$ is not fully static as the comparator is allowed to vary from a context to another. In this sense, (4) can be viewed as an intermediary between external regret and dynamic regret ([Hall and Willett](), [2013](); [Besbes et al., 2015]()). The existing literature on time-varying games typically focuses on dynamic regret ([Duvocelle et al., 2018](); [Zhang et al., 2022](); [Anagnostides et al., 2024]()), which is the most stringent notion of regret. However, the resulting bounds often include a path length term. This term captures the intrinsic variation of the comparator sequence, such as $P_T$ and $V_T$ in Example [1](). In contrast, we will show that bounds on regret (4) depend only on a prediction error term, which vanishes if agents are able to accurately predict the states of nature.

While external regret has been a cornerstone measure of performance in online learning for games, swap regret has recently aroused a lot of interest since it sets a more demanding learning benchmark and leads to tighter equilibrium concepts ([Anagnostides et al., 2022b](); [Peng and Rubinstein](), [2024](); [Dagan et al., 2024]()). We therefore supplement our analysis of regret Equation (4) by a similar study of contextual swap regret, which we introduce below. Let $\Lambda = \{\lambda : \Delta_K \to \Delta_K\}$ be the set of swap deviation maps, and define for any $j \in [J]$, $\lambda_\star^j : \Delta_K \times \mathcal{Z} \to \Delta_K$ the optimal swap comparator, which satisfies for any $z \in \mathcal{Z}$ and any $\lambda \in \Lambda$:

$$\sum_{t \in \mathcal{T}^z} c^j(\lambda_\star^j(w_t^j, z), \mathbf{w}_t^{-j}, z) \leqslant \sum_{t \in \mathcal{T}^z} c^j(\lambda(w_t^j), \mathbf{w}_t^{-j}, z) \,.$$

Given a context $z$, the swap comparator $\lambda_\star^j$ maps any played strategy $w_t^j$ to an alternative strategy that would have achieved better performance on the time steps when $z$ was observed. Importantly, competing against $\lambda_\star^j$ is strictly more demanding than competing against $\pi_\star^j$ from (3). While $\pi_\star^j$ assigns a fixed optimal action to each context $z$, independent of the strategies actually played, $\lambda_\star^j(w, z)$ adapts to the specific strategy $w$ used. This flexibility allows $\lambda_\star^j$ to identify better-performing alternatives conditioned on past decisions, making it a stronger—and thus more challenging—benchmark.

With $w_t^j = \pi_t^j(\hat{Z}_t^j)$ being the strategy played by agent $j$ at

time $t$, we define the following contextual swap regret:

$$\overline{\mathfrak{R}}_T^j = \sum_{t \in [T]} \left[ c^j(w_t^j, \mathbf{w}^{-j}, Z_t) - c^j(\lambda_\star^j(w_t^j, Z_t), \mathbf{w}_t^{-j}, Z_t) \right] .$$

(5)

Finally, in the non-contextual case, the Blum-Mansour reduction (Blum and Mansour, 2007) is a convenient procedure which allows to design a no-swap regret algorithm from any no-external regret one. A natural question is whether a similar reduction exists in our setting, that is, whether an algorithm minimizing (5) can be obtained from an algorithm minimizing (4). We answer by the positive with the following proposition.

**Proposition 1.** *Assume that player $j \in [J]$ plays an algorithm $\pi^j \in \Pi^j$ achieving $\mathfrak{R}_T^j \leqslant f(J, T, K, m)$ for some $f : \mathbb{N}_+^4 \to \mathbb{R}_+$. Then, one can design an algorithm $\overline{\pi}^j \in \Pi^j$ achieving*

$$\overline{\mathfrak{R}}_T^j \leqslant K f(J, T, K, m) .$$

The explicit procedure to design $\overline{\pi}^j$ from $\pi^j$ is described in Algorithm 2, and the proof of Proposition 1 is deferred to Appendix G. A direct consequence of Proposition 1 is that any algorithm with a guarantee on external regret (4) can be converted into another algorithm with a guarantee on swap regret (5), at the cost of an additional $K$ factor. This will be particularly useful to extend the analysis from external to swap regret. Note that more recent procedures (Dagan et al., 2024; Peng and Rubinstein, 2023) allow to deal with larger action spaces by reducing the dependence of $K$, yet at the cost of a degraded dependence on $T$.

**Equilibrium.** We consider two equilibrium concepts, which naturally relates to the two regrets previously defined. First, we focus on the classic contextual coarse-correlated equilibrium (Sessa et al., 2021; Maddux and Kamgarpour, 2024), whose definition is recalled below.

**Definition 1.** *[Sessa et al. 2021] Let $\varepsilon > 0$. An $\underline{\varepsilon\text{-contextual}}$ $\underline{\text{coarse-correlated equilibrium}}$ is a joint policy $\boldsymbol{\nu} : \mathcal{Z} \to \mathscr{P}(\mathcal{A})$ such that for any $j \in [J]$ and $\pi^j \in \Pi^j$:*

$$T^{-1} \sum_{t \in [T]} c^j(\boldsymbol{\nu}^j(Z_t), \boldsymbol{\nu}^{-j}(Z_t), Z_t)$$
$$\leqslant T^{-1} \sum_{t \in [T]} c^j(\pi^j(Z_t), \boldsymbol{\nu}^{-j}(Z_t), Z_t) + \varepsilon .$$

Note that Definition 1 extends the classic coarse correlated equilibrium concept (Foster and Vohra, 1998) to the case where the underlying state of nature changes over time. A more detailed interpretation of Definition 1 is provided in Appendix C.

While coarse-correlated equilibrium has been extensively studied, it is arguably weak in the sense that it only prevents

coarse deviations. On the other hand, correlated equilibrium (Aumann, 1987) is a tighter equilibrium concept which prevents swap deviations, see Appendix C for more discussion. We introduce below an equivalent concept adapted to our framework. In the following definition, $\varrho^j : \mathcal{A}^j \times \mathcal{Z} \to \mathcal{A}^j$ is any swap deviation function, which given an action and context $(a^j, z)$ returns an alternative action $\tilde{a}^j$.

**Definition 2.** *Let $\varepsilon > 0$. An $\underline{\varepsilon\text{-contextual correlated}}$ $\underline{\text{equilibrium}}$ is a joint policy $\overline{\boldsymbol{\nu}} : \mathcal{Z} \to \mathscr{P}(\mathcal{A})$ such that for any $j \in [J]$ and $\varrho^j : \mathcal{A}^j \times \mathcal{Z} \to \mathcal{A}^j$:*

$$T^{-1} \sum_{t \in [T]} \mathbb{E}_{\mathbf{a} \sim \overline{\boldsymbol{\nu}}(Z_t)} \left[ \langle \phi^j(\mathbf{a}), Z_t \rangle \right]$$
$$\leqslant T^{-1} \sum_{t \in [T]} \mathbb{E}_{\mathbf{a} \sim \overline{\boldsymbol{\nu}}(Z_t)} \left[ \langle \phi^j(\varrho^j(a^j, Z_t), \mathbf{a}^{-j}), Z_t \rangle \right] + \varepsilon .$$

Definition 2 extends the classic correlated equilibrium notion (Aumann, 1987) to the contextual case, by letting the swap functions $\varrho^j(\cdot, z)$ depend on the state of nature.

In the non-contextual case, there exists a well-known and powerful connection between external regret minimization and coarse-correlated equilibrium. If all players use no-external regret algorithms, their empirical average strategy is an approximate coarse correlated equilibrium. Crucially, we show that the same property holds in our contextual framework. In the following proposition, $n_z = |\mathcal{T}^z|$ denotes the number of time state $z \in \mathcal{Z}$ occurs.

**Proposition 2.** *Assume that for any $j \in [J]$, agent $j$ uses a policy $\pi^j \in \Pi^j$ incurring an external regret $\mathfrak{R}_T^j$ as in (4), and denote by $w_t^j = \pi_t^j(\hat{Z}_t^j)$ for any $t \in [T]$. Let $\hat{\boldsymbol{\nu}}_T : \mathcal{Z} \to \mathscr{P}(\mathcal{A})$ be such that for any $z \in \mathcal{Z}$,*

$$\hat{\boldsymbol{\nu}}_T(z) = \begin{cases} n_z^{-1} \sum_{t \in \mathcal{T}^z} w_t^1 \otimes \ldots \otimes w_t^J & \text{if } n_z > 0 , \\ (K^{-1}, \ldots, K^{-1}) & \text{otherwise} . \end{cases}$$

*Then, $\hat{\boldsymbol{\nu}}_T$ is an $\varepsilon$-contextual coarse correlated equilibrium with*

$$\varepsilon = \max_{j \in [J]} T^{-1} \mathfrak{R}_T^j .$$

The proof of Proposition 2 is deferred to Appendix G. It is clear from this proposition that if $\mathfrak{R}_T^j = o(T)$ for every $j \in [J]$, $\hat{\boldsymbol{\nu}}_T$ asymptotically convergences to an exact coarse-correlated equilibrium.

Moreover, the same connection exists for swap regret and correlated equilibrium in the non-contextual setting. We also retrieve this property in our contextual setting, as showed by the following proposition.

**Proposition 3.** *Assume that for any $j \in [J]$, agent $j$ uses a policy $\overline{\pi}^j \in \Pi^j$ incurring a swap regret $\overline{\mathfrak{R}}_T^j$ defined as in (5). Let $\hat{\boldsymbol{\nu}}_T : \mathcal{Z} \to \mathscr{P}(\mathcal{A})$ be defined as in Definition 1. Then, $\hat{\boldsymbol{\nu}}_T$ is an $\varepsilon$-contextual correlated equilibrium with*

$$\varepsilon = \max_{j \in [J]} T^{-1} \overline{\mathfrak{R}}_T^j .$$

The proof of this result can be found in Appendix G. It shows that if all players use algorithms incurring a sub-linear swap regret, their empirical average strategy converges to an exact correlated equilibrium. Both Proposition 2 and Proposition 3 are key to our analysis, since they convert individual regret guarantees into convergence rates to equilibrium. Hence, bounding individual regrets is our first objective.

**Social welfare.** On top of convergence to equilibrium, we study social welfare, and in particular whether no-regret strategies may result in a welfare close to the optimal one. In non-contextual games, the so-called Roughgarden smoothness condition (Roughgarden, 2015) is particularly convenient to address this question (Syrgkanis et al., 2015). This condition—which is satisfied by a wide class of games, including congestion games (Roughgarden and Tardos, 2002; Christodoulou and Koutsoupias, 2005), facility games and second price auctions (Roughgarden, 2015)—states that even when players deviate from the optimal strategy, the total cost in a game doesn't increase too much. Under this condition, it can be shown that the average social cost converges to the optimal one times the price of anarchy.

Here, we assume that our game satisfies the contextual counterpart to the Roughgarden smoothness condition.

**H3.** *There exist $\delta > 0$ and $\mu > 0$ such that for any $\mathbf{a} \in \mathcal{A}$, $\mathbf{a}_\star \in \mathcal{A}$ and $z \in \mathcal{Z}$,*

$$\sum_{j \in [J]} \langle z, \phi_j(a_\star^j, \mathbf{a}_{-j}) \rangle \leqslant \sum_{j \in [J]} [\delta \langle z, \phi_j(\mathbf{a}_\star) \rangle + \mu \langle z, \phi_j(\mathbf{a}) \rangle] .$$

It is well known that under **H3**, $\gamma = \delta(1 - \mu)^{-1}$ is an upper bound on the price of anarchy (Roughgarden, 2015). In what follows,

$$C_t(\mathbf{w}_t) = \sum_{j \in [J]} c^j(\mathbf{w}_t, Z_t) ,$$

denotes the social cost at time $t \in [T]$ and

$$C^\star = \min_{\boldsymbol{\rho}: \mathcal{Z} \to \mathcal{A}} T^{-1} \sum_{t \in [T]} \sum_{j \in [J]} c^j(\boldsymbol{\rho}(Z_t), Z_t) ,$$

the optimal average social cost in pure strategy. The following proposition shows that under **H3**, the distance between the average social cost and the optimal one is bounded by the sum of external contextual regrets.

**Proposition 4.** *Assume **H3**. Then with $\gamma = \delta(1 - \mu)^{-1}$,*

$$\frac{1}{T} \sum_{t \in [T]} C_t(\mathbf{w}_t) \leqslant \gamma C^\star + \frac{1}{(1 - \mu)T} \sum_{j \in [J]} \mathfrak{R}_T^j .$$

The proof of Proposition 4 can be found in Appendix G. In particular, when $\sum_{j \in [J]} \mathfrak{R}_T^j = o(T)$, the average social cost is guaranteed to converge to a fraction of the optimal one. Therefore, bounding $\sum_{j \in [J]} \mathfrak{R}_T^j$ will be our second objective.

## 3. Prediction-aware learning.

**Algorithm.** In the non-contextual case, the optimistic Multiplicative Weight Update (OMWU) algorithm has proven particularly effective for controlling individual and social regrets in uncoupled multiplayer games. We propose below the predictive-OMWU algorithm, abbreviated POMWU, which is an extension of OMWU to our framework. Broadly speaking, POMWU maintains one OMWU instance per context. At the beginning of each round, agents predict the context and use the corresponding OMWU to play. Once the actual state of nature has been revealed, they update the algorithm based on the cost feedback for future rounds. The pseudo-code of POMWU is displayed in Algorithm 1.

---

**Algorithm 1** Optimistic MWU with predicted contexts (POMWU) for agent $j \in [J]$.

---

1: Initialize $\rho_{z_1} = \ldots = \rho_{z_m} = (K^{-1}, \ldots, K^{-1})$ and $\Psi_{z_1} = \ldots = \Psi_{z_m} = \mathbf{0}_{d \times K}$.
2: **for** each $t \in [T]$ **do**
3:     Predict $\hat{Z}_t^j \in \mathcal{Z}$, set $M_t^j = \Psi_{\hat{Z}_t^j}$ and $g_t^j = \rho_{\hat{Z}_t^j}$.
4:     Play $w_t^j \in \Delta_K$ where for each $\ell \in \{1, \ldots, K\}$,

$$w_t^j[\ell] = \frac{g_t^j[\ell] \exp(-\eta M_t^j[\ell] \hat{Z}_t^j)}{\sum_{k \in [K]} g_t^j[k] \exp(-\eta M_t^j[k] \hat{Z}_t^j)}$$

5:     Observe $Z_t \in \mathbb{R}^d$ and $\Phi^j(\mathbf{w}_t^{-j})$.
6:     Update $\Psi_{Z_t} \leftarrow \Phi^j(\mathbf{w}_t^{-j})$
7:     Update $\rho_{Z_t} \leftarrow \rho_{Z_t} \cdot \exp(-\eta \Phi^j(\mathbf{w}_t^{-j})^\top Z_t) .$
8: **end for**

---

**RVU analysis.** The *Regret bounded by Variation in Utility* (RVU) bound (Syrgkanis et al., 2015) is an effective method for deriving guarantees regarding individual regrets in multiplayer games. Intuitively, this approach ties players' utilities to the variation in observed utilities and played strategies. The RVU ensures that players' regrets remain low when they efficiently adapt their strategy to counter the variation in payoffs. In the following key lemma, we establish a *contextual* RVU bound adapted to our framework. In what follows, we write $\mathcal{T}^z = \{t_1^z, \ldots, t_{n_z}^z\}$ for any $z \in \mathcal{Z}$, and $L_T^j = \sum_{t \in [T]} \mathbb{1}\{\hat{Z}_t^j \neq Z_t\}$ the total number of mispredictions made by agent $j \in [J]$ throughout of the game.

**Proposition 5.** *Assume **H1** and **H2**. Any $j \in [J]$ applying Algorithm 1 with learning rate $\eta > 0$ has an external regret*

*bounded as follows:*

$$\mathfrak{R}_T^j \leqslant \frac{(5 + \ln(K))L_T^j + m\ln(K)}{\eta}$$

$$+ \eta \left( \sum_{z \in \mathcal{Z}} \sum_{i \leqslant n_z} \left\| \left( \Phi^j(\mathbf{w}_{t_i^z}^{-j}) - \Phi^j(\mathbf{w}_{t_{i-1}^z}^{-j}) \right)^\top z \right\|_\infty^2 + 4L_T^j \right)$$

$$- \frac{1}{16\eta} \sum_{z \in \mathcal{Z}} \sum_{i \leqslant n_z} \left\| w_{t_i^z}^j - w_{t_{i-1}^z}^j \right\|_1^2 .$$

Contrary to the classic RVU approach (Syrgkanis et al., 2015), the bound in Proposition 5 depends on the lengths of the *context-specific* paths $\Phi^j(\mathbf{w}_{t_1^z}^{-j}), \ldots, \Phi^j(\mathbf{w}_{t_{n_z}^z}^{-j})$ and $w_{t_1^z}^j, \ldots, w_{t_{n_z}^z}^j$. The need for this new contextual RVU stems from the fact that players may mispredict states of nature at different periods, preventing the naive use of a classic RVU, see Appendix G for more details. Note that in its current form, Proposition 5 holds for any arbitrary sequence of strategies by other agents, and does not provide an explicit bound for individual regrets.

**Remark 1** (continuing from p. 3). *It is possible to quantify $L_T^j$ under **H2** when agents use an online algorithm for predicting $(Z_t)_{t \in [T]}$. Indeed, this boils down to multiclass online classification problem, for which bounds on $L_T^j$ have been established by Daniely et al. (2014). Assume that $\mathcal{G}$ has a finite Littlestone dimension $\dim_\mathscr{L}(\mathcal{G}) < \infty$ (Littlestone, 1988). In the realizable case, that is when for every $j \in [J]$, there exists $g_j^\star \in \mathcal{G}$ such that $Z_t = g_j^\star(X_t^j)$ for any $t \in [T]$, there exists an online algorithm $g_t^j : \mathcal{X} \to \mathcal{Z}$ such that $L_T^j = \sum_{t \in [T]} \mathbb{1}\{g_t^j(X_t^j) \neq Z_t\}$ satisfies:*

$$L_t^j \leqslant \dim_\mathscr{L}(\mathcal{G}) . \tag{6}$$

*In the agnostic case, denoting $L_T^{\star j} = \min_{g_j \in \mathcal{G}} \sum_{t=1}^T \mathbb{1}\{g_j(X_t^j) \neq Z_t\}$, there exists an algorithm such that*

$$L_T^j \leqslant L_T^{\star j} + \sqrt{\frac{1}{2}\dim_\mathscr{L}(\mathcal{G})T\ln(Tm)} . \tag{7}$$

*The algorithms leading to (6) and (7), namely Algorithm 3 and Algorithm 4, are both recalled in Appendix D.*

**Equilibrium.** Equipped with Proposition 5, we first focus on convergence to equilibrium. As discussed in Section 2, this only requires bounding individual regrets. The following proposition, which follows from Proposition 5, establishes this bound.

**Proposition 6.** *Define $\overline{L}_T = \max_{j \in [J]} L_T^j$ and assume **H1** and **H2**. If all agents use Algorithm 1 with a learning rate*

$\eta > 0$, *then for any $j \in [J]$:*

$$\mathfrak{R}_T^j \leqslant \frac{(5 + \ln(K))\overline{L}_T + m\ln(K)}{\eta}$$

$$+ \eta \left[ (J-1)^2 (9T\eta^2 + 4\overline{L}_T) + 4\overline{L}_T \right] .$$

*In particular if $T = \Omega(J^2\overline{L}_T)$, setting $\eta^\star = \Theta(J^{-1/2}T^{-1/4}[\ln(K)(\overline{L}_T + m)]^{1/4})$ leads to:*

$$\mathfrak{R}_T^j = \mathcal{O}\left( [\ln(K)(\overline{L}_T + m)]^{3/4}T^{1/4}J^{1/2} \right) .$$

In the realizable case of Remark 1 where $\overline{L}_T = \mathcal{O}_T(1)$, we recover the result $\mathfrak{R}_T^j = \mathcal{O}(T^{1/4}J^{1/2})$ from Syrgkanis et al. (2015). We also observe that setting the learning rate to $\eta^\star$ requires agents to know $\overline{L}_T^j$. This is reasonable if they use the same hypothesis class, since uniform bounds on $\overline{L}_T$ are known (see e.g., Remark 1).

**Remark 1** (continuing from p. 3). *Recently, collaborative and federated learning has emerged as a topic of prime importance in Machine learning (Blum et al., 2017; Kairouz et al., 2021). One may wonder whether agents sharing a common model, so $\hat{Z}_t^j = \hat{Z}_t \in \mathcal{Z}$ for any $j \in [J]$, may improve Proposition 6. Indeed, even though agents play uncoupled strategies, policies $\pi_t^j(\hat{Z}_t)$ are implicitly coordinated as they rely on a same signal. We show in Proposition 9 in Appendix G that in this case, we can drop the assumption $T = \Omega(J^2\overline{L}_T)$ and still recover the guarantee of Proposition 6 by a direct improvement of the proof. Studying the impacts of collaborative learning in games more broadly is an interesting topic for future research.*

It is now possible to obtain an explicit convergence rate to coarse-correlated equilibrium by combining Proposition 2 with Proposition 6.

**Corollary 1.** *Assume **H1**, **H2** and $T = \Omega(J^2\overline{L}_T)$. If all agents use Algorithm 1 with $\eta^\star > 0$ as defined in Proposition 6, then $\hat{\boldsymbol{\nu}}_T$ (as defined as in Proposition 2) is an $\varepsilon$-coarse correlated equilibrium, with*

$$\varepsilon = \mathcal{O}\left( [\ln(K)(\overline{L}_T + m)]^{3/4}T^{-3/4}J^{1/2} \right) .$$

In addition, it is possible to obtain a convergence rate for the more demanding contextual correlated equilibrium concept defined in Definition 2. As a matter of fact, the reduction described in Algorithm 2 allows to transform POMWU, which enjoys the guarantee on $\mathfrak{R}_T^j$ presented in Proposition 6, into an algorithm $\bar{\pi}^j$ with a guarantee on $\overline{\mathfrak{R}}_T^j$—see Proposition 1. Then, applying Proposition 3 leads to the following corollary.

**Corollary 2.** *Assume **H1**, **H2** and $T = \Omega(J^2\overline{L}_T)$. If all agents use Algorithm 1 in conjonction with Algorithm 2 and $\eta^\star > 0$ as in Proposition 6, then $\hat{\boldsymbol{\nu}}_T$ is an $\bar{\varepsilon}$-correlated equilibrium, with*

$$\bar{\varepsilon} = \mathcal{O}\left( [K\ln(K)(\overline{L}_T + m)]^{3/4}T^{-3/4}J^{1/2} \right) .$$

**Social welfare.** We now turn our attention to social welfare when agents use POMWU. As discussed in Section 2, this requires bounding the sum of regrets. A first, naive approach would be to sum the bound of individual regrets obtained in Proposition 6. However, we show below that another choice of $\eta$ leads to a much better guarantee.

**Proposition 7.** *Let $L_T = \sum_{j \in [J]} L_T^j$, and assume **H**1, **H**2. If all agents use Algorithm 1 with a learning rate $\eta = (4(J-1))^{-1}$, then*

$$\sum_{j \in [J]} \mathfrak{R}_T^j \leqslant 4J[(5 + \ln(K))L_T + mJ\ln(K)] + \frac{L_T}{J-1}$$

$$= \mathcal{O}(J\ln(K)(L_T + mJ)) \ .$$

Note that in the setting of Remark 1 under the realizable assumption, $L_T = \mathcal{O}_T(1)$ and hence we recover the classic result $\sum_{j \in [J]} \mathfrak{R}_T^j = \mathcal{O}_T(1)$ of Syrgkanis et al. (2015) in the static setting.

The bound in Proposition 7 can immediately be converted into a convergence rate of social cost to a fraction of the optimal one via Proposition 4.

**Corollary 3.** *Assume **H**1, **H**2 and **H**3. If Assume all agents use Algorithm 1 with $\eta = (4(J-1))^{-1}$, then*

$$\frac{1}{T} \sum_{t \in [T]} C_t(\mathbf{w}_t) \leqslant \gamma C^\star + \mathcal{O}(J\ln(K)T^{-1}(L_T + mJ)) \ .$$

**Robustness.** Finally, we turn our attention to the adversarial regime where not all agents use POMWU. Specifically, we ask whether the regret of POMWU remains low against any arbitrary sequence of cost feedback. This robustness property is a common desiderata in the literature (Syrgkanis et al., 2015; Foster et al., 2016).

**Proposition 8.** *Assume **H**1 and **H**2. If player $j \in [J]$ uses Algorithm 1 with $\eta = \Theta([\ln(K)(L_T^j + m)]^{1/2}(L_T^j + T)^{-1/2})$, then for any sequence $(\mathbf{w}_1^{-j}, \ldots, \mathbf{w}_T^{-j}) \in \mathscr{P}(\mathcal{A}^{-j})^T$:*

$$\mathfrak{R}_T^j = \mathcal{O}\left(\sqrt{\ln(K)(L_T^j + m)(L_T^j + T)}\right) \ .$$

Here again, in the setting of Remark 1 under realizability, $L_T^j = \mathcal{O}_T(1)$ and therefore we recover the guarantee $\mathfrak{R}_T^j = \mathcal{O}(\sqrt{T})$.

# 4. Experiments.

**Setting.** We illustrate the performances of POMWU on the Sioux Falls routing problem from LeBlanc et al. (1975) with the parameters from Sessa et al. (2019). We consider a network of cities connected by roads. In each city, there is one agent willing to send a given quantity of goods to

each other city. Agents want to minimize their travel time, which is determined by both congestion on the network, and external factors such as weather and road condition. Formally, we consider a graph $(\mathcal{V}, \mathcal{E})$ with $J = |\mathcal{V}|(|\mathcal{V}| - 1)$ agents, each of whom wants to send $q_j > 0$ units from $n_j \in \mathcal{V}$ to $m_j \in \mathcal{V}$. For any $j \in [J]$, we let $\mathcal{A}^j$ be the set of $K > 0$ shortest paths connecting $n_j$ to $m_j$, that is any $a^j \in \mathcal{A}^j$ can be written as $a^j = (i_1^j, \ldots, i_R^j)$ with $i_1^j = n_j$, $i_R^j = m_j$, and $(i_r, i_{r+1}) \in \mathcal{E}$ for any $r \in \{1, \ldots, R-1\}$. For any profile of actions $\mathbf{a} = (a^1, \ldots, a^J) \in \mathcal{A}$ and pair of nodes $(p, \ell) \in \mathcal{E}$, we denote by

$$\phi_{p,\ell}^j(\mathbf{a}) = \begin{cases} \sum_{i \in [J]} \mathbb{1}\{(p,\ell) \in a^i\}q_i^4 & \text{if } (p,\ell) \in a^j \\ 0 & \text{otherwise} , \end{cases}$$

the total congestion[1] faced by $j \in [J]$ on $(p,\ell)$, and $\phi^j(\mathbf{a}) = (\phi_{p,\ell}^j(\mathbf{a}))_{p,\ell} \in \mathbb{R}^{|\mathcal{V}| \times |\mathcal{V}|}$ the corresponding matrix. Agents are allowed to randomize over routes, so they play $w^j \in \Delta_K$. To each pair $(p,\ell) \in \mathcal{V} \times \mathcal{V}$, we also associate a cost coefficient $z_{p,\ell} > 0$ related to road condition or weather, and we denote by $Z = (z_{p,\ell})_{p,\ell} \in \mathbb{R}^{|\mathcal{V}| \times |\mathcal{V}|}$ the corresponding matrix. Then for any $\mathbf{w} \in \mathscr{P}(\mathcal{A})$ and $Z \in \mathbb{R}^{|\mathcal{V}| \times |\mathcal{V}|}$, the cost for any $j \in [J]$ is given by:

$$c^j(\mathbf{w}, Z) = \mathbb{E}_{\mathbf{w}}\left[\langle Z, \phi^j(\mathbf{a})\rangle_{\mathrm{F}}\right] \ ,$$

where $\langle A, B\rangle_{\mathrm{F}} = \mathrm{Tr}(A^\intercal B) = \sum_{i,j} A_{i,j} B_{i,j}$ is the Frobenius inner product. $c^j$ captures the expected travel time of player $j \in [J]$ when they pick routes according to $w^j \in \Delta_K$ and other agents according to $\mathbf{w}^{-j} \in \mathscr{P}(\mathcal{A}^{-j})$ under context $Z \in \mathcal{Z}$. Additional experimental details can be found in Appendix A.

**Supervised learning.** In our experiment, there are $m > 0$ random contexts denoted $\mathcal{Z} = \{z_1, \ldots, z_m\}$. For any $z \in \mathcal{Z}$, there exists $\beta_z^\star \in \mathbb{R}^b$ such that

$$\mathbb{P}(Z = z | X^0) = \zeta(\beta_z^\star, X^0) = \frac{\exp(\beta_z^\star X^0)}{\sum_{z' \in \mathcal{Z}} \exp(\beta_{z'}^\star X^0)} \ ,$$

where $X^0 \in \mathbb{R}^b$ is a vector of covariates (which can be thought of as a meteorogical or a traffic forecast) drawn from a standard Normal multivariate distribution. At each round $t \in [T]$, agents observe $X_t^0 \in \mathbb{R}^b$ and predict with a logistic regression $\hat{Z}_t \in \mathcal{Z}$, that is $\hat{Z}_t = \mathrm{argmax}_{z \in \mathcal{Z}} \zeta(\hat{\beta}_z, X_t^0)$. They then update $\hat{\beta}_{z_1}, \ldots, \hat{\beta}_{z_m}$ in an online fashion with a stochastic gradient descent. More details can be found in Appendix A.

---

[1]In Sessa et al. (2019), the congestion is of form $\tilde{\phi}_{p,\ell}(\mathbf{a}) = (\sum_{k \in [J]} \mathbb{1}\{(p,\ell) \in a^k\}q_k)^4$. We only keep the term $q_k^4$ in this sum so $\phi_{k,q}$ is linear in $\mathbf{a} \in \mathcal{A}$, which is necessary to compute expectations given the size of action space $|\mathcal{A}| = m^{|\mathcal{V}|(|\mathcal{V}|-1)}$.

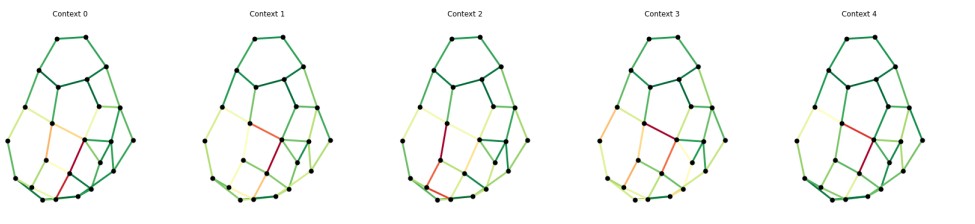

Figure 1: Average repartition of agents on the network for each context under a $10^{-3}$-coarse correlated equilibrium.

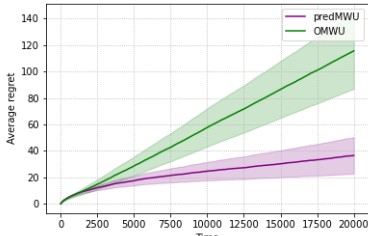

Figure 2: Average regret over agents for `POMWU` and `OMWU`.

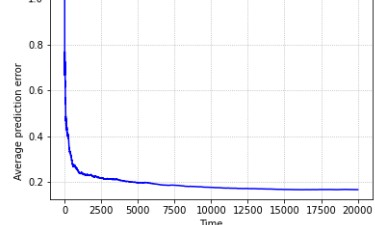

Figure 3: Average prediction error from the online logistic regression.

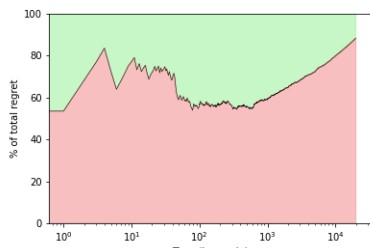

Figure 4: Proportion of average regret incurred under mispredicted contexts.

**Game.** There are $T > 0$ rounds. At each $t \in [T]$, A pair $(X_t^0, Z_t)$ is drawn, each agent $j \in [J]$ observe $X_t^0$, predict $\hat{Z}_t^j$, and play $w_t^j \in \Delta_K$ according to Algorithm 1. They then receive $Z_t$ and $(\mathbb{E}_{\mathbf{w}_t^{-j}}[\phi^j(a_{t,k}^j, \mathbf{a}_t^{-j})])_{k \in [K]}$ as a feedback, which they use to update `POMWU` and their logistic regression. The parameters used in our experiment are summarized in Appendix A.

**Results.** Figure 2 displays the the regret averaged over players[2] for a naive `OMWU` algorithm which ignores states of nature, and `POMWU`. The effectiveness of `POMWU` in adapting to time-varying payoffs is clear, especially when compared to the classic `OMWU`, whose contextual regret grows linearly due to its inability to account for states of nature. Interestingly, Figure 4 shows that rounds where contexts are mispredicted contributes to a large and growing share of regret over time for `POMWU`. This illustrates the convergence of the algorithm on each context. The fact that average prediction error of the online logistic regression (Figure 3) decreases at a slow rate thus explains most of the regret trend of `POMWU` in late rounds. Finally, Figure 1 depicts the average proportion of agents occupying each edge of the network in different contexts under the empirical policy $\hat{\nu}$ defined in Proposition 2. By Proposition 2, this is a depiction of a $10^{-3}$-approximate coarse correlated equilibrium of the game.

## 5. Conclusion

The recent extension of uncoupled learning to time-varying games marks a significant progress, as it enables the modeling of non-stationary payoff environments. However, existing literature overlooks the fact that they may be able to forecast future variations of the game. In this work, we introduce prediction-aware learning, a framework in which agents can leverage predictions about future payoffs to inform their strategies. Specifically, we propose the `POMWU` algorithm, inspired by the classic `OMWU` approach, which incorporates the predicted state of nature into the optimism step. We provide explicit guarantees on both individual regrets and social welfare, and demonstrate the effectiveness of `POMWU` in a simulated contextual game.

We believe that these findings provide a strong foundation for incorporating predictive capabilities into dynamic game-theoretic settings, with significant implications for strategic decision-making in economic and industrial applications. There are several avenues for future work to improve and expand upon this framework. First, it would be valuable to weaken the feedback provided to players—for instance, by restricting it to bandit feedback—and analyze the impact on theoretical guarantees. Second, extending the model to accommodate an infinite number of contexts presents a challenging but important direction. Finally, exploring how collaborative inference influences the game dynamics and designing algorithms that account for this interplay remains an essential question from a game-theoretic perspective.

---

[2]Shaded areas correspond to standard error computed over multiple runs.

## Impact Statement

This paper presents work whose goal is to advance the understanding of multi-agent systems. There are many potential societal consequences of our work, none which we feel must be specifically highlighted here.

## Acknowledgment

Funded by the European Union (ERC, Ocean, 101071601). Views and opinions expressed are however those of the author(s) only and do not necessarily reflect those of the European Union or the European Research Council Executive Agency. Neither the European Union nor the granting authority can be held responsible for them.

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

# A. Experiment.

**Additional information about the setting of the experiment.** The graph used to model the Sioux Falls road network from LeBlanc et al. (1975) has $|\mathcal{N}| = 24$ nodes and $|\mathcal{E}| = 76$ edges. The network topology, the cost coefficients $z_{p,\ell} > 0$ as well as the quantities $q_j > 0$ to be sent are downloaded from `https://github.com/sessap/contextualgames/tree/main/SiouxFallsNet`. In the experiment, we consider $m = 5$ states of nature. Each state of nature $i \in [m]$ is generated by adding a noise $\varepsilon_{p,\ell}^i > 0$ drawn from an exponential distribution with scale parameter $\lambda = 10^{-2}$ to each edge $(p,\ell) \in \mathcal{E}$. For each player $j \in [J]$, we let $\mathcal{A}^j$ be the $K = 5$ shortest paths connecting $n_j \in \mathcal{N}$ to $m_j \in \mathcal{N}$. While there are $|\mathcal{N}|(|\mathcal{N}| - 1) = 552$ agents in total on the network, we exclude agents for whom the lengths of the longest path exceeds the length of the shortest path by more than 2. This is because the optimal action tends to trivially be the shortest path irrespective of the state of nature for these agents. With this choice, we are left with $J = 91$ agents having actions generating rewards of the same order of magnitude. The simulation is run over $T = 2.10^4$ timesteps. The displayed regrets for `predMWU` and `OMWU` are averaged over agents.

**Online supervised learning in the experiment.** As explained in the main text, for any $t \in [T]$,

$$\mathbb{P}(Z_t = z | X_t^0) = \zeta(\beta_z^\star, X^0) = \frac{\exp(\beta_z^\star X^0)}{\sum_{z' \in \mathcal{Z}} \exp(\beta_{z'}^\star X^0)} \ ,$$

where $X^0 \in \mathbb{R}^b$ with $b = 10$. In the experiment, for any $t \in [T]$, $X_t^0 \sim \mathcal{N}(m, 5I_b)$ with $m \in [1, 4]^b$. All agents receive the same covariates from sack of simplicity. For any $z \in \mathcal{Z}$, $\beta_z^\star \in \mathbb{R}^b$ is drawn before the simulation according to a Normal distribution $\mathcal{N}(0, 5I_b)$.

# B. Discussion of H2.

In this appendix, we briefly outline two possible approaches to carry our analysis without **H**2. More precisely, we assume in this section that $\mathcal{Z} \subset$ a compact context set.

1. One first natural strategy involves discretizing $\mathcal{Z}$ using an $\varepsilon$-net and projecting each incoming context $z \in \mathcal{Z}$ to its nearest neighbor on the net. Under suitable smoothness conditions on the loss or reward functions, this could enable a reduction to the finite context case. However, this method typically results in regret bounds with exponential dependence on the dimension $d$, which can severely limit its applicability in high-dimensional settings (see, e.g., Theorem 4 in Hazan and Megiddo (2007)).

2. Alternatively, one could seek to work directly with the continuous context space. However, without placing restrictions on the policy class, it is possible to construct problem instances where both external and swap regrets grow linearly in the number of rounds. This highlights the necessity of controlling model complexity, for example by (i) restricting to a finite policy class (as in Auer et al. (2002)), or (ii) assuming linear structure in the policies (cf. LinUCB-style approaches), possibly combined with complexity measures such as sequential Rademacher complexities (Rakhlin et al., 2015). Each of these directions would require introducing new assumptions, proof techniques, and analytical frameworks, and therefore warrants a dedicated study.

# C. Discussion on Definition 1 and Definition 2.

In this section, we provide further interpretation about the equilibrium concepts defined in Definition 1 and Definition 2. First, of all, recall that $\varepsilon$-contextual coarse correlated equilibrium is defined as follows:

**Definition 1.** *[Sessa et al. 2021] Let $\varepsilon > 0$. An $\underline{\varepsilon\text{-contextual coarse-correlated equilibrium}}$ is a joint policy $\boldsymbol{\nu} : \mathcal{Z} \to \mathscr{P}(\mathcal{A})$ such that for any $j \in [J]$ and $\pi^j \in \Pi^j$:*

$$T^{-1} \sum_{t \in [T]} c^j(\nu^j(Z_t), \boldsymbol{\nu}^{-j}(Z_t), Z_t)$$
$$\leqslant T^{-1} \sum_{t \in [T]} c^j(\pi^j(Z_t), \boldsymbol{\nu}^{-j}(Z_t), Z_t) + \varepsilon \ .$$

For any $z \in \mathcal{Z}$, the distribution $\boldsymbol{\nu}(z)$ can be interpreted as a correlation device that generates and recommends pure actions to agents. We say that $\boldsymbol{\nu}(z)$ is an equilibrium in the sense of Definition 1 if no player can decrease their expected cost by ignoring the recommendations from $\boldsymbol{\nu}$ *before* they have even been drawn on average over time. Note that contrary to the classic coarse correlated equilibrium (Foster and Vohra, 1998), $\boldsymbol{\nu} : z \in \mathcal{Z} \mapsto \boldsymbol{\nu}(z) \in \mathcal{P}(\mathcal{A})$ is a *policy* that maps contexts to distributions over joint actions.

Second, a $\varepsilon$-correlated equilibrium is defined as follows.

**Definition 2.** *Let $\varepsilon > 0$. An $\varepsilon$-contextual correlated equilibrium is a joint policy $\overline{\nu} : \mathcal{Z} \to \mathscr{P}(\mathcal{A})$ such that for any $j \in [J]$ and $\varrho^j : \mathcal{A}^j \times \mathcal{Z} \to \mathcal{A}^j$:*

$$T^{-1} \sum_{t \in [T]} \mathbb{E}_{\mathbf{a} \sim \overline{\nu}(Z_t)} \big[ \big\langle \phi^j(\mathbf{a}), Z_t \big\rangle \big]$$

$$\leqslant T^{-1} \sum_{t \in [T]} \mathbb{E}_{\mathbf{a} \sim \overline{\nu}(Z_t)} \big[ \big\langle \phi^j(\varrho^j(a^j, Z_t), \mathbf{a}^{-j}), Z_t \big\rangle \big] + \varepsilon \ .$$

Just as before, $\overline{\nu}$ can be regarded as a correlation device. It is an equilibrium in the sense of Definition 2 if no player can decrease their expected cost by deviating from their recommended action *after* it has been drawn, on average over time. From this point of view, being a correlated equilibrium is more demanding than a coarse correlated equilibrium. Note that Definition 2 extends the classic correlated equilibrium notion (Aumann, 1987) to the contextual case, by letting the swap functions $\varrho^j(\,\cdot\,, z)$ depend on the state of nature.

## D. Useful algorithms.

---
**Algorithm 2** Contextual Blum-Mansour algorithm.

---
1: **Input:** a no-external regret policy $\pi^j \in \Pi^j$.
2: Initialize $h_{k,0}^j = \emptyset$ for any $k \in \{1, \dots, K\}$ .
3: **for** each $t \in \{1, \dots, T\}$: **do**
4:     Predict $\hat{Z}_t^j \in \mathcal{Z}$.
5:     For every $k \in \{1, \dots, K\}$, get $p_{k,t}^j = \pi^j(h_{k,t-1}^j, \hat{Z}_t^j)$, and define $P_t^j = (p_{1,t}^j \,|\, \dots \,|\, p_{K,t}^j) \in \mathbb{R}^{K \times K}$.
6:     Play $w_t^j \in \Delta_K$ such that
$$P_t^j w_t^j = w_t^j \ .$$
7:     Observe $Z_t \in \mathcal{Z}$ and $\Phi^j(\mathbf{w}_t^{-j}) \in \mathbb{R}^{d \times K}$ ,
8:     Update $h_{k,t}^j = h_{k,t-1}^j \cup \{w_t^j[k] \Phi^j(\mathbf{w}_t^{-j}), Z_t\}$ for ay $k \in \{1, \dots, K\}$ .
9: **end for**

---

---
**Algorithm 3** Standard Optimal Algorithm (SOA) from Daniely et al. (2014).

---
1: **Input:** An hypothesis class $\mathcal{G} \subset \{g : \mathcal{X} \to \mathcal{Z}\}$ with Littlestone dimension $\dim_{\mathscr{L}}(\mathcal{G}) < \infty$.
2: Initialize $V_0 = \mathcal{G}$.
3: **for** each $t \in \{1, \dots, T\}$: **do**
4:     Receive $X_t \in \mathcal{X}$ and define $V_t^{(z)} = \{g \in V_{t-1} : g(X_t) = z\}$ for any $z \in \mathcal{Z}$ .
5:     Predict $\hat{Z}_t \in \operatorname{argmax}_{z \in \mathcal{Z}} \dim_{\mathscr{L}}(V_t^{(z)})$ .
6:     Receive $Z_t \in \mathcal{Z}$ and update $V_t \leftarrow V_t^{(Z_t)}$ .
7: **end for**

---

## E. Notations

For the proofs, we use the following notations and shorthands.

- For any $z \in \mathcal{Z}$, $\mathscr{T}^z = \{t \in [T] : Z_t = z\} = \{t_1^z, \dots, t_{n_z}^z\}$ where $n_z = |\mathscr{T}^z|$ .

---

**Algorithm 4** Learning with Expert Advice (LEA) from Daniely et al. (2014).

---

1: **Input:** An hypothesis class $\mathcal{G} \subset \{g : \mathcal{X} \to \mathcal{Y}\}$ with Littlestone dimension $\dim_{\mathscr{L}}(\mathcal{G}) < \infty$, $N > 0$ experts using Algorithm 3 with $N \leqslant (mT)^{\dim_{\mathscr{L}}(\mathcal{G})}$.
2: Set $\eta = \sqrt{8\ln(N)/T}$ .
3: **for** each $t \in \{1, \ldots, T\}$: **do**
4:     Observe $X_t \in \mathcal{X}$, receive expert advices $(f_t^1(X_t), \ldots, f_t^N(X_t)) \in \mathcal{Z}^N$ .
5:     Predict $\hat{Z}_t = f_t^i(X_t)$ with probability proportional to $\exp(-\eta \sum_{\tau < t} \mathbb{1}\{f_\tau^i(X_\tau) \neq Z_\tau\})$.
6:     Receive $Z_t \in \mathcal{Z}$ and send it to all experts as a feedback.
7: **end for**

---

**Algorithm 5** Optimistic Mirror Descent with predicted context.

---

1: Initialize $\Psi_{z_1} = \ldots = \Psi_{z_m} = \mathbf{0}_{d \times K}$ and $\rho_{z_1} = \ldots = \rho_{z_m} = \arg\min_{\widetilde{w} \in \Delta_K} \mathcal{R}(\widetilde{w})$.
2: **for** each $t \in [T]$ **do**
3:     Observe $\hat{Z}_t^j \in \mathcal{Z}$, set $\widetilde{M}_t^j = \Psi_{\hat{Z}_t^j}$ and $\widetilde{g}_t^j = \rho_{\hat{Z}_t^j}$.
4:     Play $\widetilde{w}_t^j = \arg\min_{\widetilde{w} \in \Delta_K} \eta \left\langle \widetilde{M}_t^{j\top} \hat{Z}_t^j, \widetilde{w} \right\rangle + D_{\mathcal{R}}(\widetilde{w}, \widetilde{g}_t^j)$ ,
5:     Observe $Z_t \in \mathbb{R}^d$ and $\Phi^j(\mathbf{w}_t^{-j}) \in \mathbb{R}^{d \times K}$ .
6:     Compute $\tilde{\rho}_t = \arg\min_{g \in \Delta_K} \left\langle \Phi^j(\mathbf{w}_t^{-j})^\top Z_t, g \right\rangle + D_{\mathcal{R}}(g, \widetilde{g}_t^j)$ ,
7:     Update $\Psi_{Z_t} \leftarrow \Phi^j(\mathbf{w}_t^{-j})$ and $\rho_{Z_t} \leftarrow \tilde{\rho}_t$.
8: **end for**

---

**Algorithm 6** Optimistic FTRL with predicted context.

---

1: Initialize $w_0 = \arg\min_{w \in \Delta_K} \mathcal{R}(w)$ and $\Psi_{z_1} = \ldots = \Psi_{z_m} = \mathbf{0}_{d \times K}$.
2: **for** each $t \in [T]$ **do**
3:     Observe $\hat{Z}_t^j \in \mathcal{Z}$ and set $\widetilde{M}_t^j = \Psi_{\hat{Z}_t^j}$ .
4:     Play $w_t^j = \arg\min_{w \in \Delta_K} \left\langle \sum_{\tau=1}^{t-1} \mathbb{1}\{Z_\tau = Z_t^j\} \Phi^j(\mathbf{w}_\tau^{-j})^\top Z_\tau + \widetilde{M}_t^{j\top} Z_t^j, w \right\rangle + \frac{D_{\mathcal{R}}(w)}{\eta}$ ,
5:     Observe $Z_t \in \mathbb{R}^d$ and $\Phi^j(\mathbf{w}_t^{-j})$, update $\Psi_{Z_t} \leftarrow \Phi^j(\mathbf{w}_t^{-j})$ .
6: **end for**

---

- For any $j \in [J]$, $z \in \mathcal{Z}$ and $i \in \{1, \ldots, n_z\}$:

$$\begin{aligned}
\Phi^j(\mathbf{w}_{t_i^z}^{-j}) = \Phi_{z,i}^j \qquad &w_{t_i^z}^j = w_{z,i}^j \\
M_{t_i^z}^j = M_{z,i}^j \qquad &g_{t_i^z}^j = g_{z,i}^j \\
\tilde{\rho}_{t_i^z}^j = \tilde{\rho}_{z,i}^j \qquad &\hat{Z}_{t_i^z}^j = \hat{Z}_{z,i}^j \ .
\end{aligned}$$

## F. Technical lemmas.

**Lemma 1.** *Let $j \in [J]$, $\mathbf{w} \in \mathscr{P}(\mathcal{A})$ with $\mathbf{w} = w^j \otimes \mathbf{w}^{-j}$ and $\Phi^j(\mathbf{w}^{-j}) = (\mathbb{E}_{\mathbf{w}^{-j}}[\phi^j(a_k^j, \mathbf{a}^{-j})[\ell]])_{\ell,k} \in \mathbb{R}^{d \times K}$. We have:*

$$c^j(\mathbf{w}, Z) = \left\langle Z, \Phi^j(\mathbf{w}^{-j})w^j \right\rangle .$$

*Proof.* Let $j \in [J]$, $Z \in \mathcal{Z}$ and $\mathbf{w} \in \mathscr{P}(\mathcal{A})$ with $\mathbf{w} = w^j \otimes \mathbf{w}^{-j}$. By Fubini theorem,

$$c^j(\mathbf{w}, Z) = \mathbb{E}_{w^j} \left[ \mathbb{E}_{\mathbf{w}^{-j}} \left[ \left\langle Z, \phi^j(a^j, \mathbf{a}^{-j}) \right\rangle \right] \right] = \sum_{k=1}^K w^j[k] \, \mathbb{E}_{\mathbf{w}^{-j}} \left[ \left\langle Z, \phi_j(a_k^j, \mathbf{a}^{-j}) \right\rangle \right]$$

$$= \left\langle Z, \sum_{k=1}^K w^j[k] \mathbb{E}_{\mathbf{w}^{-j}} \left[ \phi_j(a_k^j, \mathbf{a}^{-j}) \right] \right\rangle = \left\langle Z, \Phi^j(\mathbf{w}^{-j})w^j \right\rangle .$$

$\square$

**Lemma 2.** *Let $j \in [J]$. For given sequences $(Z_1, \ldots, Z_T) \in \mathcal{Z}^T$, $(\hat{Z}_1^j, \ldots, \hat{Z}_T^j) \in \mathcal{Z}^T$ and $(\mathbf{w}_1^{-j}, \ldots, \ldots, \mathbf{w}_T^{-j}) \in \mathscr{P}(\mathcal{A}^{-j})^T$, Algorithm 1 and Algorithm 5 with $\mathcal{R} : w \mapsto \sum_{k \in [K]} w[k] \ln w[k] - w[k]$ produce the same iterates: $\widetilde{w}_t^j = w_t^j$ for any $t \in [T]$.*

*Proof.* Observe that the Bregman divergence $D_{\mathcal{R}}(w, v) = \mathcal{R}(w) - \mathcal{R}(v) - \langle \nabla \mathcal{R}(v), w - v \rangle$ generated by $\mathcal{R} : w \mapsto \sum_{k \in [K]} w[k] \ln w[k] - w[k]$ is for any $w, v \in \Delta_K$:

$$D_{\mathcal{R}}(w, v) = \mathrm{KL}(w, v) \ ,$$

where KL denotes the the Kullback-Leibler divergence. Therefore, in this proof we write KL instead of $D_{\mathcal{R}}$. Algorithm 1 produces an iterate $w_t^j \in \Delta_K$ such that for any $\ell \in [K]$:

$$w_t^j[\ell] = \frac{\exp\left[-\eta\left(M_t^j[\ell]\hat{Z}_t^j + \sum_{\tau=1}^{t-1} \mathbb{1}\{Z_\tau = \hat{Z}_t^j\}\Phi^j(\mathbf{w}_\tau^{-j})[\ell]\hat{Z}_t^j\right)\right]}{\sum_{k \in [K]} \exp\left[-\eta\left(M_t^j[k]\hat{Z}_t^j + \sum_{\tau=1}^{t-1} \mathbb{1}\{Z_\tau = \hat{Z}_t^j\}\Phi^j(\mathbf{w}_\tau^{-j})[k]\hat{Z}_t^j\right)\right]} \ . \tag{8}$$

We will show that the iterate of Algorithm 5, $\widetilde{w}_t^j = \mathrm{argmin}_{w \in \Delta_K} \eta\left\langle \widetilde{M}_t^{j\mathsf{T}}\hat{Z}_t^j, w \right\rangle + \mathrm{KL}(w, \widetilde{g}_t^j)$ is equal to (8). To this end, we define

$$\mathcal{P}_t^j = \{\tau \in \{1, \ldots, t-1\} : \ Z_\tau = \hat{Z}_t^j\} \ ,$$

and we write $\mathcal{P}_t^j = \{\tau_1, \ldots, \tau_{N_t^j}\}$ where $N_t^j = \sum_{\tau < t} \mathbb{1}\{Z_\tau = \hat{Z}_t^j\}$. We prove with a recursion that for any $r \in \{1, \ldots, N_t^j\}$, we have for any $\ell \in [K]$:

$$\tilde{g}_{\tau_r}^j[\ell] = \frac{\exp\left[-\eta \sum_{i=1}^{r-1} \Phi^j(\mathbf{w}_{\tau_i}^{-j})[\ell]\hat{Z}_t^j\right]}{\sum_{k \in [K]} \tilde{g}_{\tau_i}^j[k] \exp\left[-\eta \sum_{i=1}^{r-1} \Phi^j(\mathbf{w}_{\tau_i}^{-j})[k]\right]\hat{Z}_t^j} \ , \tag{9}$$

For $r = 1$, by definition of Algorithm 5, $\tilde{g}_{\tau_1}^j = \mathrm{argmin}_{g \in \Delta_K} \mathcal{R}(g) = m^{-1}\mathbf{1}_m$ where $\mathbf{1}_m = (1, \ldots, 1)^\mathsf{T}$, so (9) is true by the convention $\sum_{i \in \emptyset} k_i = 0$. Suppose now that (9) holds true for some $r \in \{1, \ldots, N_t^j - 1\}$. By definition,

$$\tilde{g}_{\tau_{r+1}}^j = \underset{w \in \Delta_K}{\mathrm{argmin}} \ \eta\left\langle \Phi^j(\mathbf{w}_{\tau_r}^{-j})^\mathsf{T}\hat{Z}_{\tau_r}^j, g \right\rangle + \mathrm{KL}(w, \tilde{g}_{\tau_r}^j) \ .$$

Equivalently, it is the solution to

$$\min_{g \in \mathbb{R}^m} \max_{\lambda \in \mathbb{R}} \mathcal{L}(g, \lambda) \quad \text{with} \quad \mathcal{L}(g, \lambda) = \eta\left\langle \Phi^j(\mathbf{w}_{\tau_r}^{-j})^\mathsf{T}\hat{Z}_{\tau_r}^j, g \right\rangle + \mathrm{KL}(g, \tilde{g}_{\tau_r}^j) + \lambda\left(\sum_{\ell \in [K]} g[\ell] - 1\right) \ .$$

In particular $\nabla \mathcal{L}(\tilde{g}_{\tau_{r+1}}^j, \lambda) = 0$, that is for any $\ell \in [K]$:

$$\eta\Phi^j(\mathbf{w}_{\tau_r}^{-j})[\ell]\hat{Z}_{\tau_r}^j + \ln(\tilde{g}_{\tau_{r+1}}^j[\ell]) - \ln(\tilde{g}_{\tau_r}^j[\ell]) + \lambda = 0 \quad \text{so} \quad \tilde{g}_{\tau_{r+1}}^j[\ell] = \tilde{g}_{\tau_r}^j[\ell]\exp\left(-\eta\Phi^j(\mathbf{w}_{\tau_r}^{-j})[\ell]\hat{Z}_{\tau_r}^j - \lambda\right) \ . \tag{10}$$

Using the fact that $\sum_{k \in [K]} \tilde{g}_{\tau_{r+1}}^j[k] = 1$, we obtain from (10) $\exp(\lambda) = \sum_{k \in [K]} \tilde{g}_{\tau_r}^j[k]\exp(-\eta\Phi^j(\mathbf{w}_{\tau_r}^{-j})[k]\hat{Z}_{\tau_r}^j)$, so:

$$\tilde{g}_{\tau_{r+1}}^j[\ell] = \frac{\tilde{g}_{\tau_r}^j[\ell]\exp(-\eta\Phi^j(\mathbf{w}_{\tau_r}^{-j})[\ell]\hat{Z}_{\tau_r}^j)}{\sum_{k \in [K]} \tilde{g}_{\tau_r}^j[k]\exp(-\eta\Phi^j(\mathbf{w}_{\tau_r}^{-j})[k]\hat{Z}_{\tau_r}^j)} \tag{11}$$

and using the recursion assumption establishes the result. Finally, observe that

$$\tilde{w}_t^j = \underset{w \in \Delta_K}{\mathrm{argmin}} \ \eta\left\langle \widetilde{M}_t^j\hat{Z}_t^j, w \right\rangle + \mathrm{KL}(w, \tilde{g}_t^j) \ ,$$

By the same lines of computation as previously, we obtain that for any $\ell \in [K]$:

$$\tilde{w}_t^j[\ell] = \frac{\tilde{g}_t^j \exp\left[-\eta\widetilde{M}_t^j[\ell]\hat{Z}_t^j\right]}{\sum_{k \in [K]} \tilde{g}_t^j \exp\left[-\eta\widetilde{M}_t^j[k]\hat{Z}_t^j\right]} \ ,$$

Finally, by definition of Algorithm 5, $\tilde{g}_t^j \propto \tilde{g}_{N_t^j}^j \exp(-\eta \Phi^j(\mathbf{w}_{\tau_{N_t^j}}^{-j}))$, hence by (9):

$$\tilde{w}_t^j = \frac{\exp\left[-\eta\left(\sum_{i=1}^{N_t^j} \Phi^j(\mathbf{w}_{\tau_i}^{-j})[\ell] + \widetilde{M}_t^j[\ell]\right)\hat{Z}_t^j\right]}{\sum_{k\in[K]} \tilde{g}_{\tau_i}^j[k]\exp\left[-\eta\left(\sum_{i=1}^{N_t^j} \Phi^j(\mathbf{w}_{\tau_i}^{-j})[k] + \widetilde{M}_t^j[k]\right)\hat{Z}_t^j\right]} \ ,$$

Observing that $\widetilde{M}_t^j = M_t^j$ by definition for any $t \in [T]$, we obtain the desired result. $\qquad\square$

**Lemma 3.** *For given sequences* $(Z_1, \ldots, Z_T) \in \mathcal{Z}^T$, $(\hat{Z}_1^j, \ldots, \hat{Z}_T^j) \in \mathcal{Z}^T$ *and* $(\mathbf{w}_1^{-j}, \ldots, \ldots, \mathbf{w}_T^{-j}) \in \mathscr{P}(\mathcal{A}^{-j})$, *Algorithm 1 and Algorithm 6 produce the same iterates.*

*Proof.* The proof proceeds as the one of Lemma 2: writing the first order condition of step 4 in Algorithm 6 leads to the expression (8) of the iterate of Algorithm 1. $\qquad\square$

## G. Proofs.

**Proposition 1.** *Assume that player* $j \in [J]$ *plays an algorithm* $\pi^j \in \Pi^j$ *achieving* $\mathfrak{R}_T^j \leqslant f(J, T, K, m)$ *for some* $f : \mathbb{N}_+^4 \to \mathbb{R}_+$. *Then, one can design an algorithm* $\overline{\pi}^j \in \Pi^j$ *achieving*

$$\overline{\mathfrak{R}}_T^j \leqslant K f(J, T, K, m) \ .$$

*Proof.* Let $j \in [J]$. Assume that there exists $\pi^j \in \Pi^j$ and $f : \mathbb{N}^4 \to \mathbb{R}_+$ such that the regret $\mathfrak{R}_T^j \in \mathbb{R}$ of $\pi^j$ satisfies:

$$\mathfrak{R}_T^j \leqslant f(J, T, K, m) \ . \tag{12}$$

We consider the policy $\overline{\pi}_T^j \in \Pi^j$ described in Algorithm 2. In this proof, we define for any $k \in \{1, \ldots, K\}$:

$$r_k^j = \sum_{t\in[T]} \left\langle w_t^j[k]\Phi^j(\mathbf{w}_t^{-j})^\intercal Z_t, p_{t,k}^j \right\rangle - \min_{\pi_k : \mathcal{Z} \to \Delta_K} \sum_{t\in[T]} \left\langle w_t^j[k]\Phi^j(\mathbf{w}_t^{-j})^\intercal Z_t, \pi_k(Z_t) \right\rangle \ ,$$

the regret incurred by $\overline{\pi}^j$ when fed with the histories $h_{k,1}^j, \ldots, h_{k,T}^j \in \mathcal{H}^j$. The swap-regret of $\overline{\pi}^j$ reads:

$$\overline{\mathfrak{R}}_T^j = \sum_{t\in[T]} \left\langle \Phi^j(\mathbf{w}_t^{-j}), w_t^j - \lambda_\star^j(w_t^j, Z_t) \right\rangle$$

Note by linearity of $c^j$ in $w^j \in \Delta_K$ (Lemma 1), defining $\Lambda_\star^j : z \in \mathcal{Z} \mapsto (\lambda_\star^j(a_1^j, z) | \ldots | \lambda_\star^j(a_K^j, z)) \in \{0, 1\}^{K\times K}$ allows to rewrite $\lambda_\star^j(w_t^j, Z_t) = \Lambda_\star^j(Z_t)w_t^j$, hence:

$$= \sum_{t\in[T]} \left\langle \Phi^j(\mathbf{w}_t^{-j})^\intercal Z_t, P_t^j w_t^j \right\rangle - \left\langle \Phi^j(\mathbf{w}_t^{-j})^\intercal Z_t, \Lambda_\star^j(Z_t)w_t^j \right\rangle \qquad (\text{because } w_t^j = P_t^j w_t^j)$$

$$= \sum_{t\in[T]}\left[\sum_{k\in[K]} \left\langle w_t^j[k]\,\Phi^j(\mathbf{w}_t^{-j})^\intercal Z_t, p_{k,t}^j \right\rangle - \sum_{k\in[K]} \left\langle w_t^j[k]\,\Phi^j(\mathbf{w}_t^{-j})^\intercal Z_t, \lambda_\star^j(a_k^j, Z_t) \right\rangle\right]$$

$$\leqslant \sum_{k\in[K]} r_{k,T}^j \leqslant K f(J, T, K, m) \ .$$

$\qquad\square$

**Proposition 4.** *Assume H3. Then with* $\gamma = \delta(1-\mu)^{-1}$,

$$\frac{1}{T}\sum_{t\in[T]} C_t(\mathbf{w}_t) \leqslant \gamma C^\star + \frac{1}{(1-\mu)T}\sum_{j\in[J]} \mathfrak{R}_T^j \ .$$

*Proof.* Let $\boldsymbol{\rho} : \mathcal{Z} \to \mathcal{A}$ be an optimal pure strategy policy, which satisfies $\sum_{t \in [T]} C_t(\boldsymbol{\rho}(Z_t)) = C^\star$. For any $j \in [J]$ and $z \in \mathcal{Z}$, we denote by $\boldsymbol{\delta}_\star^j(Z_t)$ the distribution which puts a mass 1 on the optimal action $\boldsymbol{\rho}(Z_t)[j]$ for agent $j \in [J]$. For any $(\mathbf{w}_1, \ldots, \mathbf{w}_T) \in \mathscr{P}(\mathcal{A})^T$, we have

$$
\begin{aligned}
\sum_{t=1}^T C_t(\mathbf{w}_t) = \sum_{j=1}^J \sum_{t=1}^T \left\langle Z_t, \Phi^j(\mathbf{w}_t^{-j}) w_t^j \right\rangle &\leqslant \sum_{j=1}^J \mathfrak{R}_T^j + \sum_{j=1}^J \sum_{t=1}^T \left\langle Z_t, \Phi^j(\mathbf{w}_t^{-j}) \boldsymbol{\delta}_\star^j(Z_t) \right\rangle \\
&= \sum_{j=1}^J \mathfrak{R}_T^j + \sum_{j=1}^J \sum_{t=1}^T \left\langle Z_t, \mathbb{E}_{\mathbf{w}_t^{-j}} \left[ \phi_j(\rho_\star^j(Z_t)[j], \mathbf{a}^{-j}) \right] \right\rangle \\
&\leqslant \sum_{j=1}^J \mathfrak{R}_T^j + \delta T C^\star + \mu \sum_{t=1}^T C_t(\mathbf{w}_t) ,
\end{aligned}
$$

where we used the $(\delta, \mu)$-smoothness assumption in the last line. Re-arranging the terms allows to conclude. $\square$

**Proposition 2.** *Assume that for any $j \in [J]$, agent $j$ uses a policy $\pi^j \in \Pi^j$ incurring an external regret $\mathfrak{R}_T^j$ as in* (4), *and denote by $w_t^j = \pi_t^j(\hat{Z}_t^j)$ for any $t \in [T]$. Let $\hat{\boldsymbol{\nu}}_T : \mathcal{Z} \to \mathscr{P}(\mathcal{A})$ be such that for any $z \in \mathcal{Z}$,*

$$
\hat{\boldsymbol{\nu}}_T(z) = \begin{cases} n_z^{-1} \sum_{t \in \mathscr{T}^z} w_t^1 \otimes \ldots \otimes w_t^J & \text{if } n_z > 0 , \\ (K^{-1}, \ldots, K^{-1}) & \text{otherwise} . \end{cases}
$$

*Then, $\hat{\boldsymbol{\nu}}_T$ is an $\varepsilon$-contextual coarse correlated equilibrium with*

$$
\varepsilon = \max_{j \in [J]} T^{-1} \mathfrak{R}_T^j .
$$

*Proof.* Let $j \in [J]$ and $\pi^j \in \Pi^j$. By definition, for any $z \in \mathcal{Z}$ such that $n_z > 0$,

$$
T^{-1} \sum_{t \in [T]} \mathbb{E}_{\hat{\boldsymbol{\nu}}_T(Z_t)} \left[ \phi^j(\mathbf{a}) \right] = T^{-1} \sum_{z \in \mathcal{Z}} \sum_{t \in \mathscr{T}^z} n_z^{-1} \sum_{t \in \mathscr{T}^z} \mathbb{E}_{\mathbf{w}_t} \left[ \phi^j(\mathbf{a}) \right] = T^{-1} \sum_{t \in [T]} \mathbb{E}_{\mathbf{w}_t} \left[ \phi^j(\mathbf{a}) \right] .
$$

This observation and Lemma 1 lead to:

$$
\begin{aligned}
T^{-1} &\sum_{t \in [T]} \left( c^j(\hat{\boldsymbol{\nu}}(Z_t), Z_t) - c^j(\pi^j(Z_t), \hat{\boldsymbol{\nu}}^{-j}(Z_t), Z_t) \right) \\
&= T^{-1} \sum_{t \in [T]} \mathbb{E}_{\hat{\boldsymbol{\nu}}_T(Z_t)} \left[ \left\langle Z_t, \phi^j(\mathbf{a}_t) \right\rangle \right] - T^{-1} \sum_{t \in [T]} \mathbb{E}_{\pi^j(Z_t) \otimes \hat{\boldsymbol{\nu}}_T^{-j}(Z_t)} \left[ \left\langle Z_t, \phi^j(a_t^j, \mathbf{a}_t^{-j}) \right\rangle \right] \\
&= T^{-1} \sum_{t \in [T]} \mathbb{E}_{\mathbf{w}_t} \left[ \left\langle Z_t, \phi^j(\mathbf{a}_t) \right\rangle \right] - T^{-1} \sum_{t \in [T]} \mathbb{E}_{\pi^j(Z_t) \otimes \mathbf{w}_t^{-j}} \left[ \left\langle Z_t, \phi^j(a_t^j, \mathbf{a}_t^{-j}) \right\rangle \right] \\
&= T^{-1} \sum_{t \in [T]} \left\langle Z_t, \Phi^j(\mathbf{w}_t^{-j}) w_t^j \right\rangle - T^{-1} \sum_{t \in [T]} \left\langle Z_t, \Phi^j(\mathbf{w}_t^{-j}) \pi^j(Z_t) \right\rangle \\
&\leqslant T^{-1} \sum_{t \in [T]} \left\langle Z_t, \Phi^j(\mathbf{w}_t^{-j}) w_t^j \right\rangle - T^{-1} \sum_{t \in [T]} \left\langle Z_t, \Phi^j(\mathbf{w}_t^{-j}) \pi_\star^j(Z_t) \right\rangle \\
&= T^{-1} \mathfrak{R}_T^j .
\end{aligned}
$$

$\square$

**Proposition 3.** *Assume that for any $j \in [J]$, agent $j$ uses a policy $\bar{\pi}^j \in \Pi^j$ incurring a swap regret $\bar{\mathfrak{R}}_T^j$ defined as in* (5). *Let $\hat{\boldsymbol{\nu}}_T : \mathcal{Z} \to \mathscr{P}(\mathcal{A})$ be defined as in Definition 1. Then, $\hat{\boldsymbol{\nu}}_T$ is an $\varepsilon$-contextual correlated equilibrium with*

$$
\varepsilon = \max_{j \in [J]} T^{-1} \bar{\mathfrak{R}}_T^j .
$$

*Proof.* Let $j \in [J]$ and $\varrho^j : \mathcal{A} \times \mathcal{Z} \to \mathcal{A}$. We have:

$$T^{-1} \sum_{t \in [T]} \mathbb{E}_{\hat{\nu}(Z_t)} \big[ \langle Z_t, \phi^j(\mathbf{a}) \rangle - \langle Z_t, \phi^j(\varrho^j(a^j, Z_t), \mathbf{a}^{-j}) \rangle \big]$$

$$= T^{-1} \sum_{z \in \mathcal{Z}} \sum_{t \in \mathcal{T}^z} \mathbb{E}_{\mathbf{w}_t} \big[ \langle z, \phi^j(\mathbf{a}) - \phi^j(\varrho^j(a^j, z), \mathbf{a}^{-j}) \rangle \big]$$

And denoting $\widetilde{\Phi}_z^j(\mathbf{w}_t^j) = (\mathbb{E}_{\mathbf{w}_t^{-j}}[\phi^j(\varrho^j(a_\ell^j, z), \mathbf{a}^{-j})[r]])_{r\ell} \in \mathbb{R}^{d \times K}$ for any $t \in \mathcal{T}^z$, by Lemma 1:

$$= T^{-1} \sum_{z \in \mathcal{Z}} \sum_{t \in \mathcal{T}^z} z^\intercal \Big( \Phi^j(\mathbf{w}_t^{-j}) - \widetilde{\Phi}_z^j(\mathbf{w}_t^{-j}) \Big) w_t^j$$

For any $z \in \mathcal{Z}$, define the matrix $B_z^j \in \{0,1\}^{K \times K}$ with coefficients $(B_z^j)_{k,\ell} = \mathbb{1}\{\varrho^j(a_k^j, z) = a_\ell^j\}$. Observe that $\widetilde{\Phi}_z^j(\mathbf{w}_t^{-j}) = \Phi^j(\mathbf{w}_t^{-j})B_z^j$, so:

$$= T^{-1} \sum_{z \in \mathcal{Z}} \sum_{t \in \mathcal{T}^z} \Big( z^\intercal \Phi^j(\mathbf{w}_t^{-j})w_t^j - z^\intercal \Phi^j(\mathbf{w}_t^{-j})B_z^j w_t^j \Big)$$

Denoting $\tilde{w}_t^j = B_z^j w_t^j \in \Delta_K$:

$$= T^{-1} \sum_{z \in \mathcal{Z}} \left( \sum_{t \in \mathcal{T}^z} z^\intercal \Phi^j(\mathbf{w}_t^{-j})w_t^j - \sum_{t \in \mathcal{T}^z} z^\intercal \Phi^j(\mathbf{w}_t^{-j})\tilde{w}_t^j \right)$$

$$\leqslant T^{-1} \sum_{z \in \mathcal{Z}} \left( \sum_{t \in \mathcal{T}^z} z^\intercal \Phi^j(\mathbf{w}_t^{-j})w_t^j - \sum_{t \in \mathcal{T}^z} z^\intercal \Phi^j(\mathbf{w}_t^{-j})\lambda_\star^j(w_t^j, z) \right)$$

$$= T^{-1} \overline{\mathfrak{R}}_T^j .$$

$\square$

**Proposition 5.** *Assume* **H1** *and* **H2** . *Any* $j \in [J]$ *applying Algorithm* 1 *with learning rate* $\eta > 0$ *has an external regret bounded as follows:*

$$\mathfrak{R}_T^j \leqslant \frac{(5 + \ln(K))L_T^j + m\ln(K)}{\eta}$$

$$+ \eta \left( \sum_{z \in \mathcal{Z}} \sum_{i \leqslant n_z} \left\| \left( \Phi^j(\mathbf{w}_{t_i^z}^{-j}) - \Phi^j(\mathbf{w}_{t_{i-1}^z}^{-j}) \right)^\intercal z \right\|_\infty^2 + 4L_T^j \right)$$

$$- \frac{1}{16\eta} \sum_{z \in \mathcal{Z}} \sum_{i \leqslant n_z} \left\| w_{t_i^z}^j - w_{t_{i-1}^z}^j \right\|_1^2 .$$

*Proof.* By Lemma 2, it is equivalent to show the result holds when players use Algorithm 5 with $\mathcal{R} : w \mapsto \sum_{\ell \in [K]} w_\ell \ln(w_\ell) - w_\ell$. We assume that this is the case for the rest of the proof. To lighten notation, we drop the tilde on $\tilde{g}$, $\tilde{M}$ and $\tilde{w}$ as compared to the pseudo-code Algorithm 5.

Let $j \in [J]$. We denote by $\ell_T^j(z) = \sum_{t \in \mathcal{T}^z} \mathbb{1}\{\hat{Z}_t^j \neq z\}$ the number of mispredictions of player $j$ on the context $z \in \mathcal{Z}$. We will prove the following inequality for any $z \in \mathcal{Z}$:

$$\sum_{t \in \mathcal{T}^z} \Big\langle \Phi^j(\mathbf{w}_t^{-j})^\intercal z, w_t^j - \pi_\star^j(z) \Big\rangle \leqslant \frac{(5 + \ln(K))\ell_T^j(z) + \ln(K)}{\eta} + \eta \left( \sum_{i=1}^{n_z} \left\| \left( \Phi_{z,i}^j - \Phi_{z,i-1}^j \right)^\intercal z \right\|_\infty^2 + 4\ell_T^j(z) \right)$$

$$- \frac{1}{8\eta} \sum_{i=1}^{n_z} \left\| w_{z,i}^j - w_{z,i-1}^j \right\|_1^2 . \tag{13}$$

Let $z \in \mathcal{Z}$. For any $t \in \mathcal{T}^z$, the instantaneous regret decomposes as:

$$\left\langle \Phi^j(\mathbf{w}_t^{-j})^\top z, w_t^j - \pi_\star^j(z) \right\rangle = \underbrace{\left\langle (\Phi^j(\mathbf{w}_t^{-j}) - M_t^j)^\top z, w_t^j - \tilde{\rho}_t \right\rangle}_{(a)} + \underbrace{\left\langle M_t^{j\top} z, w_t^j - \tilde{\rho}_t \right\rangle}_{(b)} + \underbrace{\left\langle \Phi^j(\mathbf{w}_t^{-j})^\top z, \tilde{\rho}_t - \pi_\star^j(z) \right\rangle}_{(c)},$$

(14)

where $\tilde{\rho}_t = \operatorname{argmin}_{g \in \Delta_K} \left\langle \Phi^j(\mathbf{w}_t^{-j})^\top z, g \right\rangle + D_\mathcal{R}(g, g_t^j)$ (see Algorithm 5). We bound each of these three terms. First, because $\| \cdot \|_1$ and $\| \cdot \|_\infty$ are dual,

$$(a) \leqslant \left\| (\Phi^j(\mathbf{w}_t^{-j}) - M_t^j)^\top z \right\|_\infty \left\| w_t^j - \tilde{\rho}_t \right\|_1.$$

(15)

For the second and third term, we use the following classic lemma, whose proof relies on the definition of the Bregman divergence and the first order condition.

**Lemma 4** ([Rakhlin and Sridharan (2013)](#)). *let $b \in \mathbb{R}^m$ and $c \in \mathbb{R}^m$, and define $a^\star = \operatorname{argmin}_{a \in \mathbb{R}^m} \langle a, c \rangle + D_\mathcal{R}(a, b)$. Then for any $d \in \mathbb{R}^m$,*

$$\langle c, a^\star - d \rangle \leqslant D_\mathcal{R}(d, b) - D_\mathcal{R}(d, a^\star) - D_\mathcal{R}(a^\star, b)$$

Since $w_t^j = \operatorname{argmin}_{w \in \Delta_K} \eta \left\langle M_t^{j\top} \hat{Z}_t^j, w \right\rangle + D_\mathcal{R}(w, g_t^j)$, applying Lemma 4 to (b) gives

$$(b) \leqslant \frac{1}{\eta} \left\langle M_t^{j\top}(z - \hat{Z}_t^j), w_t^j - \tilde{\rho}_t \right\rangle + \frac{1}{\eta} \left( D_\mathcal{R}(\tilde{\rho}_t, g_t^j) - D_\mathcal{R}(\tilde{\rho}_t, w_t^j) - D_\mathcal{R}(w_t^j, g_t^j) \right),$$

Observe that with $\mathcal{R}(p) = \sum_{\ell \in [K]} p_\ell \ln p_\ell - p_\ell$, we have $D_\mathcal{R}(p, q) = \mathrm{KL}(p, q)$. Hence by Pinsker's inequality,

$$\leqslant \frac{1}{\eta} \left\langle M_t^{j\top}(z - \hat{Z}_t^j), w_t^j - \tilde{\rho}_t \right\rangle + \frac{1}{\eta} \left( D_\mathcal{R}(\tilde{\rho}_t, g_t^j) - \frac{1}{2} \left( \left\| \tilde{\rho}_t - w_t^j \right\|_1^2 + \left\| w_t^j - g_t^j \right\|_1^2 \right) \right).$$

(16)

Likewise, $\tilde{\rho}_t = \operatorname{argmin}_{g \in \Delta_K} \left\langle \Phi_t^{(j)\top} Z_t, g \right\rangle + D_\mathcal{R}(g, g_t^j)$ so by Lemma 4:

$$(c) \leqslant \frac{1}{\eta} \left( D_\mathcal{R}(\pi_\star^j(z), g_t^j) - D_\mathcal{R}(\pi_\star^j(z), \tilde{\rho}_t) - D_\mathcal{R}(\tilde{\rho}_t, g_t^j) \right),$$

(17)

Plugging (15), (16), (17) in (14) and summing over $\mathcal{T}^z$ yields

$$\sum_{t \in \mathcal{T}^z} \left\langle \Phi^j(\mathbf{w}_t^{-j})^\top z, w_t^j - \pi_\star^j(z) \right\rangle \leqslant \sum_{t \in \mathcal{T}^z} \left\| (\Phi^j(\mathbf{w}_t^{-j}) - M_t^j)^\top z \right\|_\infty \left\| w_t^j - \tilde{\rho}_t \right\|_1 - \frac{1}{2\eta} \sum_{t \in \mathcal{T}^z} \left( \left\| w_t^j - \tilde{\rho}_t \right\|_1^2 + \left\| w_t^j - g_t^j \right\|_1^2 \right)$$

$$+ \frac{1}{\eta} \sum_{t \in \mathcal{T}^z} \left\langle M_t^{j\top}(z - \hat{Z}_t^j), w_t^j - \tilde{\rho}_t \right\rangle + \frac{1}{\eta} \sum_{t \in \mathcal{T}^z} (D_\mathcal{R}(\pi_\star^j(z), g_t^j) - D_\mathcal{R}(\pi_\star^j(z), \tilde{\rho}_t))$$

Now, since $\|a\|_\infty \|b\|_1 \leqslant \frac{\mu}{2} \|a\|_\infty^2 + \frac{1}{2\mu} \|b\|_1^2$ for any $\mu > 0$,

$$\leqslant \frac{\mu}{2} \sum_{t \in \mathcal{T}^z} \left\| (\Phi^j(\mathbf{w}_t^{-j}) - M_t^j)^\top z \right\|_\infty^2 - \left( \frac{1}{2\eta} - \frac{1}{2\mu} \right) \sum_{t \in \mathcal{T}^z} \left\| w_t^j - \tilde{\rho}_t \right\|_1^2 - \frac{1}{2\eta} \sum_{t \in \mathcal{T}^z} \left\| w_t^j - g_t^j \right\|_1^2$$

$$+ \frac{1}{\eta} \sum_{t \in \mathcal{T}^z} \left\langle M_t^{j\top}(z - \hat{Z}_t^j), w_t^j - \tilde{\rho}_t \right\rangle + \frac{1}{\eta} \sum_{t \in \mathcal{T}^z} D_\mathcal{R}(\pi_\star^j(z), g_t^j) - D_\mathcal{R}(\pi_\star^j(z), \tilde{\rho}_t) \quad (18)$$

Setting $\mu = 2\eta$ and noticing that $-1/2\eta < -1/4\eta$ leads to

$$\leqslant \underbrace{\eta \sum_{t \in \mathcal{T}^z} \left\| (\Phi^j(\mathbf{w}_t^{-j}) - M_t^j)^\top z \right\|_\infty^2}_{(i)} \underbrace{- \frac{1}{4\eta} \sum_{t \in \mathcal{T}^z} \left( \left\| w_t^j - \tilde{\rho}_t \right\|_1^2 + \left\| w_t^j - g_t^j \right\|_1^2 \right)}_{(ii)}$$

$$\underbrace{+ \frac{1}{\eta} \sum_{t \in \mathcal{T}^z} D_\mathcal{R}(\pi_\star^j(z), g_t^j) - D_\mathcal{R}(\pi_\star^j(z), \tilde{\rho}_t)}_{(iii)} \underbrace{+ \frac{1}{\eta} \sum_{t \in \mathcal{T}^z} \left\langle M_t^{j\top}(z - \hat{Z}_t^j), w_t^j - \tilde{\rho}_t \right\rangle}_{(iv)}. \quad (19)$$

We now bound each sum. First for term (i), writing $\mathcal{T}^z = \{t_1^z, \ldots, t_{n_z}^z\}$ (and using the shorthands defined in Appendix E):

$$\sum_{t \in \mathcal{T}^z} \left\| \left( \Phi^j(\mathbf{w}_t^{-j}) - M_t^j \right)^\top z \right\|_\infty^2 = \sum_{i=1}^{n_z} \left\| \left( \Phi_{z,i}^j - M_{z,i}^j \right)^\top z \right\|_\infty^2 ,$$

By definition of Algorithm 5 for any $i \in \{1, \ldots, n_z\}$ we have $M_{z,i}^j = \Phi_{z,i-1}^j$ if $\hat{Z}_{z,i}^j = z$, so:

$$= \sum_{i=1}^{n_z} \mathbb{1}\{\hat{Z}_{z,i}^j = z\} \left\| \left( \Phi_{z,i}^j - \Phi_{z,i-1}^j \right)^\top z \right\|_\infty^2 + \sum_{i=1}^{n_z} \mathbb{1}\{\hat{Z}_{z,i}^j \neq z\} \left\| \left( \Phi_{z,i}^j - M_{z,i}^j \right)^\top z \right\|_\infty^2$$

By **H**1:

$$\leqslant \sum_{i=1}^{n_z} \mathbb{1}\{\hat{Z}_{z,i}^j = z\} \left\| \left( \Phi_{z,i}^j - \Phi_{z,i-1}^j \right)^\top z \right\|_\infty^2 + 4 \sum_{i=1}^{n_z} \mathbb{1}\{\hat{Z}_{z,i}^j \neq z\}$$

$$\leqslant \sum_{i=1}^{n_z} \left\| \left( \Phi_{z,i}^j - \Phi_{z,i-1}^j \right)^\top z \right\|_\infty^2 + 4\ell_T^j(z) . \tag{20}$$

For the term (ii), observe that for any $i \in \{1, \ldots, n_z\}$,

$$\left\| w_{z,i}^j - w_{z,i-1}^j \right\|_1^2 \leqslant 4 \left\| w_{z,i}^j - g_{z,i}^j \right\|_1^2 + 4 \left\| g_{z,i}^j - \tilde{\rho}_{z,i-1}^j \right\|_1^2 + 4 \left\| w_{z,i-1}^j - \tilde{\rho}_{z,i-1}^j \right\|_1^2 . \tag{21}$$

We then have:

$$\sum_{t \in \mathcal{T}^z} \left( \left\| w_t^j - \tilde{\rho}_t \right\|_1^2 + \left\| w_t^j - g_t^j \right\|_1^2 \right) = \sum_{i=1}^{n_z} \left( \left\| w_{z,i}^j - \tilde{\rho}_{z,i} \right\|_1^2 + \left\| w_{z,i}^j - g_{z,i}^j \right\|_1^2 \right)$$

$$= \sum_{i=1}^{n_z} \left( \left\| w_{z,i-1}^j - \tilde{\rho}_{z,i-1} \right\|_1^2 + \left\| w_{z,i-1}^j - g_{z,i-1}^j \right\|_1^2 \right)$$

$$+ \underbrace{\left( \left\| w_{z,n_z}^j - \tilde{\rho}_{z,n_z} \right\|_1^2 - \left\| w_{z,0}^j - \tilde{\rho}_{z,0} \right\|_1^2 \right)}_{\geqslant 0}$$

$$\geqslant \sum_{i=1}^{n_z} \left\| w_{z,i}^j - w_{z,i-1}^j \right\|_1^2 - \left\| g_{z,i}^j - \tilde{\rho}_{z,i-1}^j \right\|_1^2 \qquad \text{(by (21))}$$

Moreover, by definition of Algorithm 5, $g_{z,i}^j = \tilde{\rho}_{z,i-1}^j$ whenever $\hat{Z}_{z,i}^j = z$, so:

$$\geqslant \frac{1}{4} \sum_{i=1}^{n_z} \left\| w_{z,i}^j - w_{z,i-1}^j \right\|_1^2 - \sum_{i=1}^{n_z} \mathbb{1}\{\hat{Z}_{z,i}^j \neq z\} \left\| g_{z,i}^j - \tilde{\rho}_{z,i-1}^j \right\|_1^2$$

$$\geqslant \frac{1}{4} \sum_{i=1}^{n_z} \left\| w_{z,i}^j - w_{z,i-1}^j \right\|_1^2 - 4\ell_T^j(z) \tag{22}$$

Regarding the term (iii), we can use the same reasoning by writing for any $i \in \{1, \ldots, n_z\}$:

$$D_\mathcal{R}(\pi_\star^j(z), g_{z,i}^j) - D_\mathcal{R}(\pi_\star^j(z), \tilde{\rho}_{z,i}^j) = D_\mathcal{R}(\pi_\star^j(z), g_{z,i}^j) - D_\mathcal{R}(\pi_\star^j(z), \tilde{\rho}_{z,i-1}^j)$$
$$+ D_\mathcal{R}(\pi_\star^j(z), \tilde{\rho}_{z,i-1}^j) - D_\mathcal{R}(\pi_\star^j(z), \tilde{\rho}_{z,i}^j) ,$$

Since $g_{z,i}^j = \tilde{\rho}_{z,i-1}^j$ if $\hat{Z}_{z,i}^j = z$, summing over $\mathcal{T}^z$ gives:

$$\sum_{i=1}^{n_z} D_\mathcal{R}(\pi_\star^j(z), g_{z,i}^j) - D_\mathcal{R}(\pi_\star^j(z), \tilde{\rho}_{z,i}^j) = \sum_{i=1}^{n_z} \mathbb{1}\{\hat{Z}_{z,i}^j \neq z\} \Big( D_\mathcal{R}(\pi_\star^j(z), g_{z,i}^j) - D_\mathcal{R}(\pi_\star^j(z), \tilde{\rho}_{z,i-1}^j) \Big)$$

$$+ \sum_{i=1}^{n_z} D_\mathcal{R}(\pi_\star^j(z), \tilde{\rho}_{z,i-1}^j) - D_\mathcal{R}(\pi_\star^j(z), \tilde{\rho}_{z,i}^j) ,$$

Observing that $0 \leqslant D_{\mathcal{R}}(p, q) \leqslant \ln(K)$ for any $(p, q) \in \Delta_K^2$ and that the second sum is telescoping:

$$\sum_{i=1}^{n_z} D_{\mathcal{R}}(\pi_\star^j(z), g_{z,i}^j) - D_{\mathcal{R}}(\pi_\star^j(z), \tilde{\rho}_{z,i}^j) \leqslant (\ell_T^j(z) + 1) \ln(K) . \tag{23}$$

Finally for the term (iv), observe that for any $t \in \mathscr{T}^z$ we have by **H**1:

$$\left\langle M_t^{j\top}(z - \hat{Z}_t^j), w_t^j - \tilde{\rho}_t \right\rangle \leqslant 4 \mathbb{1}\{\hat{Z}_t^j \neq z\} \quad \text{so} \quad \sum_{t \in \mathscr{T}^z} \left\langle M_t^{j\top}(z - \hat{Z}_t^j), w_t^j - \tilde{\rho}_t \right\rangle \leqslant 4 \ell_t^j(z) . \tag{24}$$

Then, plugging (20), (22), (23) and (24) in (19) establishes Equation (13), and summing (13) over $\mathcal{Z}$ gives the desired result. $\qquad \square$

**Proposition 7.** *Let* $L_T = \sum_{j \in [J]} L_T^j$, *and assume* **H**1, **H**2. *If all agents use Algorithm* 1 *with a learning rate* $\eta = (4(J-1))^{-1}$, *then*

$$\sum_{j \in [J]} \mathfrak{R}_T^j \leqslant 4J[(5 + \ln(K))L_T + mJ \ln(K)] + \frac{L_T}{J-1}$$

$$= \mathcal{O}(J \ln(K)(L_T + mJ)) .$$

*Proof.* Our proof follows from Syrgkanis et al. (2015) with our new RVU bound. Let $(t, t') \in [T]^2$ and $j \in [J]$. Observe that:

$$\left\| \left( \Phi^j(\mathbf{w}_t^{-j}) - \Phi^j(\mathbf{w}_{t'}^{-j}) \right)^\top z \right\|_\infty = \max_{\ell \in [K]} \left| \mathbb{E}_{\mathbf{w}_t^{-j}} \left[ \left\langle \phi_j(a_\ell, \mathbf{a}_t^{-j}), z \right\rangle \right] - \mathbb{E}_{\mathbf{w}_{t'}^{-j}} \left[ \left\langle \phi_j(a_\ell, \mathbf{a}_{t'}^{-j}), z \right\rangle \right] \right|$$

And since $\langle \phi_j(\mathbf{a}), z \rangle \leqslant 1$ for any $\mathbf{a} \in \mathcal{A}$ by **H**1, with TV denoting the total variation:

$$\leqslant \text{TV}(\mathbf{w}_t^{-j}, \mathbf{w}_{t'}^{-j}) = \text{TV}\left( \bigotimes_{k \neq j} w_t^k, \bigotimes_{k \neq j} w_{t'}^k \right)$$

$$\leqslant \sum_{k \neq j} \text{TV}(w_t^k, w_{t'}^k) = \sum_{k \neq j} \left\| w_t^k - w_{t'}^k \right\|_1 .$$

Squaring the previous inequality and applying Cauchy-Schwarz leads to

$$\left\| \left( \Phi^j(\mathbf{w}_t^{-j}) - \Phi^j(\mathbf{w}_{t'}^{-j}) \right)^\top z \right\|_\infty^2 \leqslant \left( \sum_{k \neq j} \left\| w_t^k - w_{t'}^k \right\|_1 \right)^2 \leqslant (J-1) \sum_{k \neq j} \left\| w_t^k - w_{t'}^k \right\|_1^2 , \tag{25}$$

This implies:

$$\sum_{j \in [J]} \left\| \left( \Phi^j(\mathbf{w}_t^{-j}) - \Phi^j(\mathbf{w}_{t'}^{-j}) \right)^\top z \right\|_\infty^2 \leqslant (J-1) \sum_{j \in [J]} \sum_{k \neq j} \left\| w_t^k - w_{t'}^k \right\|_1^2 = (J-1)^2 \sum_{j \in [J]} \left\| w_t^j - w_{t'}^j \right\|_1^2 . \tag{26}$$

On the other hand, summing the RVU bounds featured in Proposition 5 over players gives:

$$\sum_{j \in [J]} \mathfrak{R}_T^j \leqslant \frac{(5 + \ln(K))L_T + mJ \ln(K)}{\eta} + \eta \left( \sum_{z \in \mathcal{Z}} \sum_{i \in [n_z]} \sum_{j \in [J]} \left\| \left( \Phi_{z,i}^j - \Phi_{z,i-1}^j \right)^\top z \right\|_\infty^2 + 4L_T \right)$$

$$- \frac{1}{16\eta} \sum_{z \in \mathcal{Z}} \sum_{i \in [n_z]} \sum_{j \in [J]} \left\| w_{z,i}^j - w_{z,i-1}^j \right\|_1^2$$

Plugging (26) for any $z \in \mathcal{Z}$, $t = t_i^z$ and $t' = t_{i-1}^z$ for $i \in \{1, \dots, n_z\}$ gives:

$$\leqslant \frac{(5 + \ln(K))L_T + mJ \ln(K)}{\eta} + 4\eta L_T + \left( \eta(J-1)^2 - \frac{1}{16\eta} \right) \sum_{z \in \mathcal{Z}} \sum_{i \in [n_z]} \sum_{j \in [J]} \left\| w_{z,i}^j - w_{z,i-1}^j \right\|_1^2$$

Then, picking $\eta = (4(J-1))^{-1}$ yields the desired result. $\qquad \square$

**Lemma 5.** *If player $j \in [J]$ uses Algorithm 1 with a learning rate $\eta > 0$, for any $i \in \{1, \ldots, n_z\}$:*

$$\left\| w_{z,i}^j - w_{z,i-1}^j \right\|_1 \leqslant 3\eta \mathbb{1}\{\hat{Z}_t^j = z\} + 2(1 - \mathbb{1}\{\hat{Z}_t^j = z\}) .$$

*Proof.* Let $j \in [J]$ and $i \in \{1, \ldots, n_z\}$. By Lemma 3, it is sufficient to prove that the claim holds true for Algorithm 6. First if $\hat{Z}_{z,i}^j \neq z$, $\|w_{z,i}^j - w_{z,i-1}^j\|_1 \leqslant 2$. Second, assume that $\hat{Z}_{z,i}^j = z$. We define for any $i' \in \{1, \ldots, n_z\}$ $f_{i'} : w \mapsto \left\langle w, \sum_{r=1}^{i'-1} \Phi_{z,r}^{j\top} z + M_{z,i'}^{j\top} z \right\rangle + \eta^{-1} \mathcal{R}(w)$ and $g_{i'} : w \mapsto \left\langle w, \sum_{r=1}^{i'} \Phi_{z,r}^{j\top} z \right\rangle + \eta^{-1} \mathcal{R}(w)$. Observe that for any $w \in \Delta_K$,

$$f_i(w) - g_i(w) = \left\langle w, (M_{z,i}^j - \Phi_{z,i}^j)^\top z \right\rangle \quad \text{and} \quad f_i(w) - g_{i-1}(w) = \left\langle w, M_{z,i}^{j\top} z \right\rangle . \tag{27}$$

We also define $v_{i-1} = \operatorname{argmin}_{v \in \Delta_K} g_{i-1}(v)$. We have:

$$\left\| w_{z,i}^j - w_{z,i-1}^j \right\|_1 \leqslant \left\| w_{z,i}^j - v_{i-1} \right\|_1 + \left\| v_{i-1} - w_{z,i-1}^j \right\|_1 . \tag{28}$$

One the one hand, by $\eta^{-1}$-strong convexity of $f_i$ with respect to $\| \cdot \|_1$, we have

$$\frac{1}{2\eta} \left\| w_{z,i}^j - v_{i-1} \right\|_1 \leqslant f_i(v_{i-1}) - f_i(w_{z,i}^j) + \left\langle \nabla f_i(w_{z,i}^j), w_{z,i}^j - v_{i-1} \right\rangle$$

And since $w_{z,i}^j = \operatorname{argmin}_{w \in \Delta_K} f_i(w)$ by definition in Algorithm 6, the first order condition gives:

$$\frac{1}{2\eta} \left\| w_{z,i}^j - v_{i-1} \right\|_1 \leqslant f_i(v_{i-1}) - f_i(w_{z,i}^j) \tag{29}$$

Since $v_{i-1} = \operatorname{argmin}_{v \in \Delta_K} g_{i-1}(v)$, we obtain by the same reasoning,

$$\frac{1}{2\eta} \left\| w_{z,i}^j - v_{i-1} \right\|_1 \leqslant g_{i-1}(w_{z,i}^j) - g_{i-1}(v_{i-1}) . \tag{30}$$

Summing (29) with (30) and applying remark (27) leads to:

$$\left\| w_{z,i}^j - v_{i-1} \right\|_1^2 \leqslant \eta \left\langle v_{i-1} - w_{z,i}^j, M_{z,i}^{j\top} z \right\rangle \leqslant \eta \left\| w_{z,i}^j - v_{i-1} \right\|_1 \left\| M_{z,i}^{j\top} z \right\|_\infty$$

Dividing on both sides by $\|w_{z,i}^j - v_{i-1}\|_1$ gives:

$$\left\| w_{z,i}^j - v_{i-1} \right\|_1 \leqslant \eta \left\| M_{z,i}^{j\top} z \right\|_\infty \leqslant \eta . \tag{31}$$

Similarly, it is easy to check that

$$\frac{1}{2\eta} \left\| v_{i-1} - w_{z,i-1}^j \right\|_1^2 \leqslant f_{i-1}(v_{i-1}) - f_{i-1}(w_{z,i-1}^j) \quad \text{and} \quad \frac{1}{2\eta} \left\| w_{z,i-1}^j - v_{i-1} \right\|_1^2 \leqslant g_{i-1}(w_{z,i-1}^j) - g_{i-1}(v_{i-1}) .$$

So once again summing these two inequalities and making use of remark (27) leads to

$$\left\| w_{z,i-1}^j - v_{i-1} \right\|_1^2 \leqslant \eta \left\langle v_{i-1} - w_{z,i-1}^j, (M_{z,i}^j - \Phi_{z,i}^j)^\top z \right\rangle \leqslant \eta \left\| w_{z,i-1}^j - v_{i-1} \right\|_1 \left\| (M_{z,i}^j - \Phi_{z,i}^j)^\top z \right\|_\infty ,$$

Dividing both sides by $\left\| w_{z,i-1}^j - v_{i-1} \right\|_1$ gives:

$$\left\| w_{z,i-1}^j - v_{i-1} \right\|_1 \leqslant \eta \left\| (M_{z,i}^j - \Phi_{z,i}^j)^\top z \right\|_\infty \leqslant 2\eta . \tag{32}$$

Finally, plugging (31) and (32) in (28) yields the result. $\qquad \square$

**Proposition 6.** *Define $\overline{L}_T = \max_{j \in [J]} L_T^j$ and assume **H**1 and **H**2. If all agents use Algorithm 1 with a learning rate $\eta > 0$, then for any $j \in [J]$:*

$$\mathfrak{R}_T^j \leqslant \frac{(5 + \ln(K))\overline{L}_T + m\ln(K)}{\eta}$$
$$+ \eta\big[(J-1)^2(9T\eta^2 + 4\overline{L}_T) + 4\overline{L}_T\big] \,.$$

*In particular if $T = \Omega(J^2\overline{L}_T)$, setting $\eta^\star = \Theta(J^{-1/2}T^{-1/4}[\ln(K)(\overline{L}_T + m)]^{1/4})$ leads to:*

$$\mathfrak{R}_T^j = \mathcal{O}\big(\,[\ln(K)(\overline{L}_T + m)]^{3/4}T^{1/4}J^{1/2}\,\big) \,.$$

*Proof.* Let $j \in [J]$. By Proposition 5 we know that

$$\mathfrak{R}_T^j \leqslant \frac{(5 + \ln(K))L_T^j + m\ln(K)}{\eta} + \eta\left(\sum_{z \in \mathcal{Z}}\sum_{i \in [n_z]}\left\|\big(\Phi_{z,i}^j - \Phi_{z,i-1}^j\big)^\mathsf{T}z\right\|_\infty^2 + 4L_T^j\right) \,. \tag{33}$$

Moreover, we proved in (25) that for any $z \in \mathcal{Z}$ and $i \in \{1, \ldots, n_z\}$, $\|(\Phi_{z,i}^j - \Phi_{z,i-1}^j)^\mathsf{T}z\|_\infty^2 \leqslant (J-1)\sum_{k \neq j}\|w_{z,i}^k - w_{z,i-1}^k\|_1^2$, so summing over contexts and timesteps gives:

$$\sum_{z \in \mathcal{Z}}\sum_{i \in [n_z]}\left\|(\Phi_{z,i}^j - \Phi_{z,i-1}^j)^\mathsf{T}z\right\|_\infty^2 \leqslant (J-1)\sum_{z \in \mathcal{Z}}\sum_{i \in [n_z]}\left(\sum_{k \neq j}\|w_{z,i}^k - w_{z,i-1}^k\|_1^2\right)$$

Applying Lemma 5 yields:

$$\leqslant (J-1)\sum_{z \in \mathcal{Z}}\sum_{i \in [n_z]}\sum_{k \neq j}\Big(3\eta\mathbb{1}\{\hat{Z}_{z,i}^k = z\} + 2\mathbb{1}\{\hat{Z}_{z,i}^k \neq z\}\Big)^2$$

$$= (J-1)\sum_{z \in \mathcal{Z}}\sum_{k \neq j}\left(\sum_{i \in [n_z]}9\eta^2\mathbb{1}\{\hat{Z}_{z,i}^k = z\} + 4\mathbb{1}\{\hat{Z}_{z,i}^k \neq z\}\right)$$

$$\leqslant (J-1)\sum_{k \neq j}\left(\sum_{z \in \mathcal{Z}}9n_z\eta^2 + 4\ell_T^k(z)\right)$$

Therefore,

$$\sum_{z \in \mathcal{Z}}\sum_{i \in [n_z]}\left\|(\Phi_{z,i}^j - \Phi_{z,i-1}^j)^\mathsf{T}z\right\|_\infty^2 \leqslant (J-1)\sum_{k \neq j}(9T\eta^2 + 4L_T^k) \leqslant (J-1)^2(9T\eta^2 + 4\overline{L}_T) \,. \tag{34}$$

Plugging (34) into (33) establishes the first part of the proposition. For the second part of the proposition, define for any $\eta > 0$:

$$h(\eta) = \frac{(5 + \ln(K))\overline{L}_T + m\ln(K)}{\eta} + \eta\big[(J-1)^2(9T\eta^2 + 4\overline{L}_T) + 4\overline{L}_T\big]$$

$$= \frac{a}{\eta} + b\eta^3 + c\eta \quad \text{with} \quad \begin{cases} a = (5 + \ln(K))\overline{L}_T + m\ln(K) \\ b = 9(J-1)^2T \\ c = 4[(J-1)^2 + 1]\overline{L}_T \,. \end{cases}$$

We are looking for a minimizer of $h$ to make the bound tight. Since $h$ is continuous and $\lim_{\eta \to 0} h(\eta) = \lim_{\eta \to \infty} h(\eta) = +\infty$, it admits a minimum on $(0, \infty)$, which is also unique by strict convexity. By the first order condition, $h$ is minimized for

$$\eta^\star = \sqrt{\frac{\sqrt{12ab + c^2} - c}{6b}} \,.$$

We now determine the order of magnitude of $\eta^\star$. On the one hand, by sub-additivity of $x \mapsto \sqrt{x}$:

$$\eta^\star \leqslant (12ab)^{1/4}(6b)^{-1/2} = \mathcal{O}((ab)^{1/4}b^{-1/2}) \, . \tag{35}$$

On the other hand, observe that by assumption $T = \Omega(J^2 \overline{L}_T)$, so $ab = \Omega(c^2)$. Consequently, for $T > 0$ large enough there exists $\gamma > 0$ such that $12ab \geqslant \gamma c^2$ and it follows that

$$\sqrt{c^2 + 12ab} - c = \int_{c^2}^{c^2 + 12ab} \frac{dt}{2\sqrt{t}} \geqslant \frac{12ab}{2\sqrt{c^2 + 12ab}} \geqslant \frac{12ab}{2\sqrt{1 + \gamma^{-1}}\sqrt{12ab}} \geqslant \frac{\sqrt{12ab}}{2\sqrt{1 + \gamma^{-1}}} \, ,$$

so we deduce that for $T > 0$ large enough,

$$\eta^\star \geqslant \sqrt{\frac{\sqrt{12ab}}{12\sqrt{1 + \gamma^{-1}}b}} \quad \text{that is} \quad \eta^\star = \Omega((ab)^{1/4}b^{-1/2}) \, .$$

Therefore, $\eta^\star = \Theta((ab)^{1/4}b^{-1/2})$. Plugging this value in $h$ finally gives:

$$h(\eta^\star) = \Theta(b^{1/4}a^{3/4} + a^{1/4}cb^{-1/4}) = \mathcal{O}(b^{1/4}a^{3/4}) \, ,$$

because $c = \mathcal{O}(a^{1/2}b^{1/2})$ by assumption. Replacing $a$ and $b$ with their actual values yields the claimed bound. □

**Proposition 8.** *Assume **H**1 and **H**2. If player $j \in [J]$ uses Algorithm 1 with $\eta = \Theta([\ln(K)(L_T^j + m)]^{1/2}(L_T^j + T)^{-1/2})$, then for any sequence $(\mathbf{w}_1^{-j}, \ldots, \mathbf{w}_T^{-j}) \in \mathscr{P}(\mathcal{A}^{-j})^T$:*

$$\mathfrak{R}_T^j = \mathcal{O}\left( \sqrt{\ln(K)(L_T^j + m)(L_T^j + T)} \right) \, .$$

*Proof.* Let $j \in [J]$ and $(\mathbf{w}_1^{-j}, \ldots, \mathbf{w}_T^{-j}) \in \mathscr{P}(\mathcal{A}^{-j})^T$ be any sequence of competitor strategies. We have for any $(t, t') \in [T]^2$ and $z \in \mathcal{Z}$:

$$\left\| \left( \Phi^j(\mathbf{w}_t^{-j}) - \Phi^j(\mathbf{w}_{t'}^{-j}) \right)^\top z \right\|_\infty^2 \leqslant 2\left\| \Phi^j(\mathbf{w}_t^{-j})^\top z \right\|_\infty^2 + 2\left\| \Phi^j(\mathbf{w}_{t'}^{-j})^\top z \right\|_\infty^2$$

$$\leqslant 2\left( \max_{k \in [d]} \left\langle \mathbb{E}_{\mathbf{w}_t^{-j}}\left[ \phi^j(a_\ell^j, \mathbf{a}^{-j}) \right], z \right\rangle \right)^2 + 2\left( \max_{k \in [d]} \left\langle \mathbb{E}_{\mathbf{w}_{t'}^{-j}}\left[ \phi^j(a_\ell^j, \mathbf{a}^{-j}) \right], z \right\rangle \right)^2$$

$$= 2\left( \max_{k \in [d]} \mathbb{E}_{\mathbf{w}_t^{-j}}\left[ \left\langle \phi^j(a_\ell^j, \mathbf{a}^{-j}), z \right\rangle \right] \right)^2 + 2\left( \max_{k \in [d]} \mathbb{E}_{\mathbf{w}_{t'}^{-j}}\left[ \left\langle \phi^j(a_\ell^j, \mathbf{a}^{-j}), z \right\rangle \right] \right)^2$$

$$\leqslant 4 \, ,$$

where we have used **H**1 in the last line. Therefore by Proposition 5, we have:

$$\mathfrak{R}_T^j \leqslant \frac{(5 + \ln(K))L_T^j + m\ln(K)}{\eta} + 4\eta(T + L_T^j) = \mathcal{O}\left( \frac{\ln(K)(L_T^j + m)}{\eta} + \eta(L_T^j + T) \right) \, .$$

Then, setting $\eta = \Theta([\ln(K)(L_T^j + m)]^{1/2}(L_T^j + T)^{-1/2})$ leads to

$$\mathfrak{R}_T^j = \mathcal{O}([\ln(K)(L_T^j + m)]^{j1/2}(L_T^j + T)^{1/2}) \, .$$

□

**Proposition 9.** *Suppose that for any $t \in [T]$, there exists $\hat{Z}_t \in \mathcal{Z}$ such that $\hat{Z}_t^j = \hat{Z}_t$ for any $j \in [J]$, and let $\underline{L}_T = \sum_{t \in [T]} \mathbb{1}\{\hat{Z}_t \neq Z_t\}$. Assume **H**1 and **H**2. If all agents use Algorithm 1 with a learning rate $\eta^\star = \Theta(J^{-1/2}T^{-1/4}[\ln(K)(\underline{L}_T + m)]^{1/4})$, then:*

$$\mathfrak{R}_T^j = \mathcal{O}\left( [\ln(K)(\underline{L}_T + m)]^{3/4}T^{1/4}J^{1/2} \right) \, .$$

*Proof.* In this proof, we write for any $z \in \mathcal{Z}$ and $i \in \{1, \ldots, n_z\}$, $\hat{Z}_{t_i^z} = \hat{Z}_{z,i}$. By Proposition 5 we know that

$$\mathfrak{R}_T^j \leqslant \frac{(5 + \ln(K))\underline{L}_T + m\ln(K)}{\eta} + \eta\left(\sum_{z \in \mathcal{Z}} \sum_{i=1}^{n_z} \left\|\left(\Phi_{z,i}^j - \Phi_{z,i-1}^j\right)^\mathsf{T} z\right\|_\infty^2 + 4\underline{L}_T\right). \tag{36}$$

For any $z \in \mathcal{Z}$, we define $\underline{\ell}_T(z) = \sum_{t \in \mathcal{T}^z} \mathbb{1}\{\hat{Z}_t \neq z\}$ and $\mathscr{C}^z = \{i \in \{1, \ldots, n_z\} : \hat{Z}_{z,i} = z \text{ and } \hat{Z}_{z,i-1} = z\}$. We have:

$$\sum_{z \in \mathcal{Z}} \sum_{i=1}^{n_z} \left\|\left(\Phi_{z,i}^j - \Phi_{z,i-1}^j\right)^\mathsf{T} z\right\|_\infty^2 = \sum_{z \in \mathcal{Z}} \left(\sum_{i \in \mathscr{C}^z} \left\|\left(\Phi_{z,i}^j - \Phi_{z,i-1}^j\right)^\mathsf{T} z\right\|_\infty^2 + \sum_{i \notin \mathscr{C}^z} \left\|\left(\Phi_{z,i}^j - \Phi_{z,i-1}^j\right)^\mathsf{T} z\right\|_\infty^2\right)$$

Note that $\mathcal{T}^z \setminus \mathscr{C}^z = \{i \in \{1, \ldots, n_z\} : \hat{Z}_{z,i} \neq z \text{ or } \hat{Z}_{z,i-1} \neq z\}$ so $|\mathcal{T}^z \setminus \mathscr{C}^z| \leqslant 2\underline{\ell}_T(z)$. Together with the fact that $\|(\Phi_{z,i}^j - \Phi_{z,i-1}^j)^\mathsf{T} z\| \leqslant 4$ for any $j \in [J]$ and $i \in \mathcal{T}^z \setminus \mathscr{C}^z$, this implies:

$$\leqslant \sum_{z \in \mathcal{Z}} \left(\sum_{i \in \mathscr{C}^z} \left\|\left(\Phi_{z,i}^j - \Phi_{z,i-1}^j\right)^\mathsf{T} z\right\|_\infty^2 + 8\underline{\ell}_T(z)\right)$$

We proved in (25) that for any $z \in \mathcal{Z}$ and $i \in \{1, \ldots, n_z\}$, $\|(\Phi_{z,i}^j - \Phi_{z,i-1}^j)^\mathsf{T} z\|_\infty^2 \leqslant (J-1)\sum_{k \neq j} \|w_{z,i}^k - w_{z,i-1}^k\|_1^2$, so

$$\leqslant \sum_{z \in \mathcal{Z}} \left((J-1) \sum_{t \in \mathscr{C}^z} \sum_{k \neq j} \left\|w_{z,i}^k - w_{z,i-1}^k\right\|_1^2 + 8\underline{\ell}_T(z)\right)$$

And by Lemma 5:

$$\leqslant \sum_{z \in \mathcal{Z}} \left(9(J-1)^2 |\mathscr{C}^z| \eta^2 + 8\underline{\ell}_T(z)\right)$$

$$\leqslant 9(J-1)^2 T\eta^2 + 8\underline{L}_T .$$

Plugging this bound in (36) yields:

$$\mathfrak{R}_T^j \leqslant \tilde{h}(\eta) = \frac{\tilde{a}}{\eta} + \tilde{b}\eta^3 + \tilde{c}\eta \quad \text{with} \quad \begin{cases} \tilde{a} &= (5 + \ln(K))\underline{L}_T + m\ln(K) \\ \tilde{b} &= 9(J-1)^2 T \\ \tilde{c} &= 12\underline{L}_T . \end{cases}$$

We observe that $\tilde{c} \propto J^{-2}c$, where $c > 0$ is defined in the proof of Proposition 6. In particular, since $\underline{L}_T \leqslant T$, we have $\tilde{a}\tilde{b} = \Omega(\tilde{c}^2)$, hence we do not need it as an assumption. The rest of the proof follows exactly as in Proposition 6. $\qquad\square$

