# OpenReview forum: "Prediction-Aware Learning in Multi-Agent Systems"
_ICML.cc/2025/Conference — ICML 2025 poster_

### Official Review · Reviewer_DNiF · 2025-03-01

**Overall Recommendation:** 3

**Summary:**

The paper considers learning in time-varying multi-player games using prediction-aware algorithms. It begins by observing that prior results quickly become vacuous when there is a large variation between the games, even though the underlying sequence can be entirely predictable. In light of this, the paper proposes a new algorithm that incorporates predictions under the framework of contextual learning, wherein, in each round, the algorithm makes a prediction about the underlying state of nature. When the sequence of states is somewhat predictable. the paper shows that existing results from the static setting can be generalized even when there is substantial variation between the games.

**Claims And Evidence:**

All claims made in the paper are clear and sound.

**Essential References Not Discussed:**

I did not identify any essential reference missing.

**Experimental Designs Or Analyses:**

The experimental design and the interpretation of the results are, as far as I can ascertain, sound.

**Methods And Evaluation Criteria:**

The paper is mostly theoretical. The experimental evaluation supports the theoretical claims and drives home the main argument of the paper: it shows that existing algorithms, namely OMWU, fail to take into account the predictability in the underlying sequence of states, unlike the proposed algorithm. The sequence of games constructed is sufficiently rich to make a convincing argument--it's not just a toy example.

**Other Comments Or Suggestions:**

Some minor points:

- What is ARIMA in Line 130? I might have missed the definition.

- there is a typo in "is an interesting topics for future research" in Line 348.

**Other Strengths And Weaknesses:**

On the positive side, the paper makes a clear contribution by addressing an important gap in prior work in the context of time-varying games. The proposed algorithm is natural and very much relevant in practice--as long as the number of states is reasonably small. I can certainly see many possible applications in which the results of the paper can be used. The paper is also very well-written and organized. The key ideas of the paper are explained very clearly.

One drawback is that most of the results are not particularly challenging to obtain, and follow mostly by adapting in a straightforward way existing techniques. But I do not believe that this alone is a basis for rejection.

**Questions For Authors:**

The paper mostly follows the techniques of Syrgkanis et al. I wonder whether the authors tried to use some more recent results that, in the static setting, obtain polylog(T) regret bounds. I also did not find any guarantees about POMWU instantiated with Blum-Mansour, which is introduced in the preliminaries; but I might have missed that.

**Relation To Broader Scientific Literature:**

The contributions of the paper lie mostly in the area of learning in games, and in particular, time-varying games. It addresses some drawbacks of existing results, as I pointed out above. The paper does a good job at placing the contributions accurately in the context of existing work. In particular, it extends the framework of Syrgkanis et al. from the static setting to the time-varying setting.

**Theoretical Claims:**

I checked the proofs, and all claims appear to be sound; I did not find any notable issue.

---

> ### Author Rebuttal · Authors · 2025-03-28
>
> We thank the reviewer for their very relevant remarks. We answer below to the points they raise.
>
> > “What is ARIMA in Line 130? I might have missed the definition.”
>
> ARIMA refers to Auto Regressive Integrated Moving Average process, a popular process in time series analysis (see, e.g., [3]). We will make this acronym explicit in the revised version of the text.
>
> [3] Hamilton, J. D. (2020). Time series analysis. Princeton university press.
>
> > “there is a typo in "is an interesting topics for future research" in Line 348.”
>
> Thank you for pointing out this typo, we will correct it in the revised version of the text.
>
> > “The paper mostly follows the techniques of Syrgkanis et al. I wonder whether the authors tried to use some more recent results that, in the static setting, obtain polylog(T) regret bounds. [...].”
>
> We appreciate the reviewer’s insightful question. We did consider leveraging techniques that achieve polylog(T) regret guarantees in the static setting, such as those in [4] and [5]. However, several limitations led us to build upon Syrgkanis’ framework instead.
>
> Regarding [4], the proof techniques are particularly intricate, and it remains unclear whether their approach can be extended to the contextual setting. In fact, the complexity of [4]’s analysis is explicitly acknowledged in the abstract of [5].
>
> As for [5], its theoretical framework—based on self-concordant barriers—appears more adaptable to prediction-aware learning. However, it comes with several drawbacks. First, the polylog(T) regret bound in [5] (Corollary 4.5) introduces a dependence on the number of players J and the number of pure actions K through a factor of JK^{5/2}. In contrast, our external and swap regret bounds scales as $\sqrt{J}  \ln K$ and $ \sqrt{J} K \ln K$ (Propositions 1 and 6) respectively, which is significantly more favorable.
> Additionally, [5] is somewhat less general in scope, as its analysis is restricted to swap regret and does not apply to external regret. This limitation arises because their main proof relies on the positivity of swap regret, which is not guaranteed for external regret. Consequently, adopting their approach would prevent us from providing guarantees for coarse correlated equilibria and social welfare, as Roughgarden’s smoothness condition relies on external regret.
>
> [4] Daskalakis, C., Fishelson, M., & Golowich, N. (2021). Near-optimal no-regret learning in general games. Advances in Neural Information Processing Systems, 34, 27604-27616.
>
> [5] Anagnostides, I., Farina, G., Kroer, C., Lee, C. W., Luo, H., & Sandholm, T. (2022). Uncoupled learning dynamics with $ o (\log t) $ swap regret in multiplayer games. Advances in Neural Information Processing Systems, 35, 3292-3304.
>
> > " [...] I also did not find any guarantees about POMWU instantiated with Blum-Mansour, which is introduced in the preliminaries; but I might have missed that."
>
> The Blum-Mansour reduction from external to swap regret stated in Proposition 1 is used to establish Corollary 2-(ii). We will highlight this more clearly in the text of the revised version.

---

### Official Review · Reviewer_W7RW · 2025-03-07

**Overall Recommendation:** 4

**Summary:**

This paper considers time-varying games where better guarantees (wrt (swap) regret, equilibrium concepts and social welfare)  can be achieved when the agents can predict/estimate the time-varying utilities. For a J-player game with utility $c^{j}(w,Z)$ (where $Z$ captures the time-variance), the players use a variant of optimistic exponential weights method to predict $\hat Z_t$ and update the iterates $w_t$. The analysis is relying on the assumption that $Z$ is from a finite set, which limits the variations of the time-varying games.
The authors provide guarantees for the regret of each individual player (Prop. 4) which is of similar order as the bound for optimistic Hedge plus a multiplicative factor of $\bar L_T$ (total number of mispredictions of $Z$). From this a bound on the social welfare is derived (Prop 5) and guarantees for the approximation quality of an $\epsilon$-CCE and $\epsilon$-CE. Robustness guarantees wrt the incentive of deviating from the proposed strategy are provided in Prop. 7. Furthermore, the theoretical findings are supplemented with illustrating experiments.

**Claims And Evidence:**

Yes.

**Essential References Not Discussed:**

The authors should consider adding the reference 'Multi-agent online learning in time-varying games', Benoit Duvocelle, Panayotis Mertikopoulos, Mathias Staudigl, Dries Vermeulen, in addition to the reference to Zhang et al.

**Ethical Review Concerns:**

none.

**Experimental Designs Or Analyses:**

This is theoretical work. The illustrating experiments make sense.

**Methods And Evaluation Criteria:**

This is theoretical work. The illustrating experiments make sense.

**Other Comments Or Suggestions:**

none.

**Other Strengths And Weaknesses:**

The paper is clearly written, nice to read and provides an interesting extension to existing results.

**Questions For Authors:**

none.

**Relation To Broader Scientific Literature:**

The work extends the research on time-varying games in the case where the time-variance can be estimated. It relates to the research on time-varying games on the one hand and ideas from contextual bandits (however, full feedback is considered here!) on the other hand.

**Theoretical Claims:**

Yes, mostly (for anything that was not clear from the claims). To the best of my understanding, all theoretical results are correct.

---

> ### Author Rebuttal · Authors · 2025-03-28
>
> We sincerely appreciate the reviewer’s positive feedback and the highly relevant reference they have suggested. We will incorporate it into our literature review in the revised version.

---

### Official Review · Reviewer_798F · 2025-03-08

**Overall Recommendation:** 2

**Summary:**

This paper discusses the problem of no-regret learning in general time-varying games with predictions. The authors argue that current regrets, defined as a function of variations in the payoff matrix and variations in the Nash equilibria, become vacuous even in simple examples like the one provided in Example 1. In this way, the paper motivates an alternative viewpoint toward prediction-aware learning by considering prediction oracles that account for changes in nature (parameterized in some way). Building on this, the authors define the notions of external and swap contextual regret and generalize the constructions of [Blum and Mansour, 2007], RVU bounds, CCE convergence, and social welfare guarantees of [Syrgkanis et al., 2015] for this new setting.

**Claims And Evidence:**

Yes, the theoretical claims are well supported; the claims are fairly standard, and there is no magic.

**Essential References Not Discussed:**

None that I can think of.

**Experimental Designs Or Analyses:**

The routing problem used is quite interesting, and the results seem valid.

**Methods And Evaluation Criteria:**

The authors evaluate the proposed algorithm on the Sioux Falls routing problem from [LeBlanc et al. (1975)] to assess its performance under mispredicted contexts.

**Other Comments Or Suggestions:**

I cannot suggest acceptance of this work at this stage since it seems too incremental. However, I am willing to reconsider if the prediction-aware rates under regression oracles yielding regret rates matching those of [Syrgkanis et al., 2015], with perfect predictions as a special case, are included in the results.

**Other Strengths And Weaknesses:**

The significance of the results is questionable. Once the problem is well-defined in terms of costs, etc., the derivations are straightforward and follow standard techniques in the literature.

The authors start the motivation interestingly with an example of predictions that are regression-based, but the focus of the paper is solely on online classification and Littlestone dimension, postponing the more challenging and interesting problem to future work. This work, despite strongly motivated examples, seems incomplete.

**Questions For Authors:**

No questions. The paper is well-written and straightforward.

**Relation To Broader Scientific Literature:**

Learning in time-changing games with supervised learning is an interesting and fundamental problem with many real-world applications.

**Theoretical Claims:**

Even though I skimmed, I don’t think any major issues exist since the theoretical claims are pretty standard.

---

> ### Author Rebuttal · Authors · 2025-03-28
>
> We thank the reviewer for their feedback. We reply to the points they raise below.
>
> > “The significance of the results is questionable. Once the problem is well-defined in terms of costs, etc., the derivations are straightforward and follow standard techniques in the literature.”
>
> We respectfully disagree with the characterization of our work as incremental. Our contributions go beyond standard derivations in several important ways.
>
> First, we view the formalization of the problem — including the explicit definition of costs — as a significant contribution in itself. In doing so, we introduce several novel concepts, such as contextual swap regret and contextual correlated equilibrium, which represent meaningful advances in the study of contextual multi-player decision-making.
>
> Second, the derivation of our results are not as straightforward as they may seem at first sight. Beyond handling the inherent complexity of contextual decision-making, we must also account for the intricate dynamics introduced by multiple players. In particular, players may predict different contexts at each timestep, preventing a direct application of Syrgkanis’ analysis on batches of timesteps with the same revealed context. To address this challenge, we develop a novel regret decomposition in the proof of proposition 4, which is essential for our analysis and enables us to reach our conclusions.
>
> > “The authors start the motivation interestingly with an example of predictions that are regression-based [...]”
>
> We believe our motivating example aligns with the classification setting, as it involves only two distinct payoff matrices (one for even timesteps, one for odd timesteps), which can be mapped to a {0,1} context set.
>
> > “[...]  postponing the more challenging and interesting problem to future work. This work, despite strongly motivated examples, seems incomplete.”
>
> While many real-world problems can be modeled with a finite number of contexts, we acknowledge the reviewer’s point that extending our framework to regression is both valuable and important. However, such an extension would require an entirely different set of analytical tools and falls beyond the scope of this paper.

---

> > ### Comment · Reviewer_798F · 2025-04-01
> >
> > Thanks a lot for your detailed response. While I acknowledge that this problem is interesting, my concern regarding the restrictiveness of the online classification settings remains unaddressed.
> >
> > The motivation in Example 1 of the manuscript that criticizes the measures of variations of
> > $$ P\_T = \min\_{\mathcal{E}\_1 \times \ldots \times \mathcal{E}\_T}   \sum\_{t \in [T]}  \left( \|\| x\_t^\star - x\_{t-1}^\star \| \|\_1 + \|\| y\_t^\star - y\_{t-1}^\star \|\|\_1 \right), $$
> > and
> > $$
> > V_T = \sum_{t \in [T]} \|\| A_t - A_{t-1} \|\|_\infty^2,
> > $$
> > used in regret bounds by (Zhang et al., 2022) and (Anagnostides et al., 2024) is very strong, but the results of this paper being only limited to cases when variations are limited to a finite number of classes instead of a regression problem is a little bit disappointing and incremental. For this reason, I am not changing my decision, but I am open to reconsidering significantly (as mentioned already in my review) if additional results, matching those of [Syrgkanis et al., 2015], with perfect predictions as a special case, are included.

---

> > > ### Author Response · Authors · 2025-04-04
> > >
> > > We sincerely thank the reviewer for their response. Even though many applications we envision feature categorical contexts, we would be happy to explore the suggested extension. However, we would like to reiterate that addressing it would require an entirely different set of tools and analysis; and thus warrants a separate study. Below, we outline the key reasons for this conclusion.
> > >
> > > Assume that $\mathcal{Z}$ is now a compact subset of $\mathbb{R}^d$. Two approaches are possible.
> > >
> > > First, we could discretize $\mathcal{Z}$, project any context $z\in\mathcal{Z}$ onto the closest node of the resulting mesh, and then apply our analysis. However, this would lead to an exponential dependence on the dimension.
> > >
> > > Second, we could work directly with a continuum of contexts. However, without restrictions on the policy class, it is always possible to construct an instance in which our notions of contextual and swap regrets grow linearly with $T$. To address this, we would need to either introduce new regret concepts or consider specific policy classes whose complexity can be controlled, such as linear policies. In both cases, this would involve intricate analysis requiring substantial time and effort, making it difficult to incorporate into the current paper, which already introduces numerous concepts and results.
> > >
> > > Finally, we note that the reviewer acknowledges that our paper is well-motivated, addresses a novel problem, and provides an effective solution both empirically and theoretically under our finite context set assumption. While we recognize that the suggested extension is both interesting and promising, we hope we have convinced the reviewer that it falls outside the scope of the present study.

---

### Official Review · Reviewer_RGHK · 2025-03-13

**Overall Recommendation:** 3

**Summary:**

This paper introduces a prediction-aware learning framework for time-varying games, where agents can forecast future payoffs and adapt their strategies accordingly. The authors propose the POWMU algorithm, a contextual extension of the optimistic Multiplicative Weight Update algorithm, and provide theoretical guarantees on social welfare and convergence to equilibrium. The framework achieves performance comparable to static settings when prediction errors are bounded

**Claims And Evidence:**

1. Introduction of a prediction-aware learning framework for time-varying games
2. Development of the POWMU algorithm for leveraging predictions about the state of nature
3. Theoretical guarantees on social welfare and convergence to equilibrium are the evidences for the paper.
4. Empirical demonstration of POWMU's effectiveness in a traffic routing experiment also varifies the claim.

**Essential References Not Discussed:**

NA

**Experimental Designs Or Analyses:**

Empirical demonstration of POWMU's effectiveness in a traffic routing experiment is shown, however, this alone might not be sufficient, I would like to see some more experiments.

**Methods And Evaluation Criteria:**

1.The major method and the evaluation criteria used is regret, and the gurantees on the social welfare.

**Other Comments Or Suggestions:**

NA

**Other Strengths And Weaknesses:**

Strenghts:

1. Addresses a gap in existing literature by incorporating agents' predictive capabilities
2. Provides theoretical guarantees that match non-contextual guarantees for static games under certain conditions
3. Demonstrates robustness in adversarial settings

Weaknesses:

1. Assumes bounded prediction errors, which may not always be realistic
2. Limited empirical evaluation (only one traffic routing experiment)
3. Complexity of the proposed algorithm might not be practical to implement.

**Questions For Authors:**

1. How does the performance of POWMU compare to other state-of-the-art algorithms in time-varying games?
2. Can the framework be extended to handle partial information or bandit feedback settings?
3. How sensitive is the algorithm to the quality of predictions, and what happens when prediction errors are not bounded?
4. Are there any specific real-world applications where this approach could provide significant improvements over existing methods?
5. How does the complexity of the proposed algorithm handled?

**Relation To Broader Scientific Literature:**

The problem is well posed and novel, it will be a good addition to the missing literature.

**Theoretical Claims:**

The proofs are well written and I have skimmed through the proofs, though not verified them line by line, but they appear to be correct.

---

> ### Author Rebuttal · Authors · 2025-03-28
>
> We thank the reviewer for their valuable feedback. We respond below to the weaknesses they indicate, and reply to their questions.
>
> > “The paper assumes bounded prediction errors, which may not always be realistic.”
>
> We emphasize that our results hold in full generality, without requiring a bounded number of mispredictions. The case where $L_T​=O(1)$ is presented solely for illustrative purposes, highlighting the connection with the well-known static guarantees from Syrgkanis (2015). However, this is not an assumption underlying our analysis.
>
> > “Limited empirical evaluation  [...]”
>
> We thought that our traffic routing experiment was sufficient to demonstrate the empirical effectiveness of POWMU, which seems to be an opinion shared by other reviewers. However, we are open to adding another experiment if the reviewer considers it helpful. An interesting case would be optimal trading strategy on a financial market, where thehe actual price of the asset would depend (for instance linearly) on the actions of the traders. Before making orders on the market, the trader would forecast the future prices of the asset based on public information. We could work with the historical financial data and contextual information from [3] to run the experiment.
>
> [3] Wang et al., 2023. Robust Contextual Portfolio Optimization with Gaussian Mixture Models.
>
> > “Complexity of the proposed algorithm[...].”
> If the reviewer refers to the algorithm's design complexity, we respectfully emphasize that POWMU is quite straightforward to implement. Indeed, the method merely involves maintaining a separate instance of OMWU for each context and updating these instances based on feedback from nature. If the reviewer's concern pertains to computational or memory complexity, we also note that POWMU is not particularly demanding in this respect. The algorithm primarily requires storing vectors and matrices and performing only basic operations, such as element-wise multiplication, matrix multiplication, and exponentiation—all of which are computationally efficient.
>
> > “How does the performance of POWMU compare to other state-of-the-art algorithms in time-varying games?”
>
> The most recent studies on time-varying games [1, 2] primarily use OMWU as their main algorithm. Figure 2 in our paper compares the regret incurred by OMWU and POWMU in our experiments, showing that POWMU significantly outperforms OMWU. This advantage arises because POWMU leverages context predictions, whereas OMWU relies solely on the previous round’s information to determine its strategy. When the game dynamics change gradually, OMWU can perform well, as suggested by the regret bounds in [1] and [2]. However, in highly unstable payoff environments, our approach achieves substantially better performance, as it selectively incorporates information from past rounds with similar contexts.
>
> [1] Zhang et al., 2022. No-regret learning in time-varying zero-sum games.
> [2] Anagnostides et al. 2023. On the convergence of no-regret learning dynamics in time-varying games.
>
> > Can the framework be extended to handle partial information or bandit feedback settings?
> We believe extending prediction-aware learning to partial information or bandit feedback is a promising yet challenging direction. A potential approach could involve adapting the Low Approximate Regret (LAR) concept from [3] to the contextual setting, building upon standard bandit algorithms to create a bandit variant of POWMU. However, working with the LAR property is inherently more complex than handling RVU, making such an adaptation non-trivial. Although this extension lies beyond the scope of the current paper, we recognize it as an important avenue for future research..
>
> > How sensitive is the algorithm to the quality of predictions,[...]
>
> Propositions 5 and 6 explicitly characterize the dependence of the social and individual regret bounds on prediction errors. Specifically, our bound for social welfare grows linearly with $L_T$, while individual regret scales as $\bar{L}_T^{3/4}$. Regarding the boundedness of prediction errors, we kindly refer the reviewer to the first point of our response for further clarification.
>
> > "Are there any specific real-world applications where this approach could provide significant improvements over existing methods?"
>
> Our framework is broad enough to be applied to various real-world scenarios. Examples include power production and trading, where the state of nature may encompass factors such as wind speed and temperature, which influence renewable energy generation and overall power consumption. Other applications include portfolio management, where agents predict future asset prices, and traffic management. We are currently engaging with a power production firm to explore the application of our method to power generation and trading.
>
> > “How does the complexity of the proposed algorithm handled?”
>
> We kindly refer the reviewer to the third point of our reply for this question.

---

### Decision · Program_Chairs · 2025-05-01

**Decision:**

Accept (poster)

**Comment:**

This paper discusses the impact of predictability in time-varying games. Specifically, the authors consider a contextual normal form game in mixed strategies where the cost to player $i$ is of the form
$$
c_j(x;z)
	= \sum_{a_1\in\mathcal A_1} \dotsb \sum_{a_J\in\mathcal A_J}
	x_{1,a_1}\dotsm x_{J,a_J} \, c_j(a_1,\dots,a_N;z)
$$
where $z$ is a context variable, the $a$'s represent pure strategies, the $x$'s are mixed strategies, and the inner $c_j$ is linear in $z$. During play, the context variable $z$ evolves over a finite set, leading to a time-varying game. The authors then provide bounds for a contextual version of the external and swap regret of the players, assuming that they all follow a likewise contextual version of the optimistic multiplicative weights algorithm. Finally, assuming the game satisfies a certain "contextual smoothness" condition (a context-based version of Roughgarden's $(\lambda,\mu)$-smoothness condition), they translate these bounds into social welfare approximation guarantees.

The paper's main premise is the following observation: previous results on time-varying games become vacuous when there is high variation in the underlying game, even when this variation is completely predictable. This is a solid and insightful observation, which was well-received by the reviewers. At the same time however, the paper is not without its flaws:
1. The paper is heavy on notation, often dense and not always well-motivated, and which significantly hampers readability—even for experts. 1. Assumptions are introduced without sufficient justification, including the finiteness of the context set (a point of concern for reviewers), and the “linear bandit” structure. This lack of clarity makes it difficult to parse the mathematical content and evaluate the plausibility of the results.[1. For example, the smoothness assumption H3 is said to hold for a wide class of games, but the only examples provided are context-free.]
2. The motivation for several modeling choices is often underdeveloped. For example, it is unclear why the authors emphasize swap regret rather than the dynamic regret - which is more commonly studied in time-varying games. Choices of this type feel incidental – as opposed to driven by a clear conceptual argument – making the narrative of the paper weaker.
3. The technical insights in the paper are also relatively limited: for example, in the case of the swap regret mentioned above, the authors invoke the standard Blum-Mansour reduction, and there is little more going on than that, so it is not clear how this expands our understanding (and likewise for the RVU framework of Syrgkanis et al., etc.).
4. The paper also has some issues with positioning - for instance, given the work of Besbes et al (2015) on dynamic regret, or the original paper on time-varying games by Duvocelle et al (2018).

Overall, it was not easy to make a recommendation for this paper: the authors tackle an important problem with a compelling starting point, but the presentation needs substantial revision. The writing should be simplified, the assumptions clarified, and the technical and conceptual motivations better developed.

My honest opinion is that the paper does not quite clear the bar for acceptance in its current form; at the same time however, it is not out of line with other papers that have appeared in ICML (or similar venues). I am therefore making a "conditional accept" recommendation contingent on a significant rewrite that addresses the concerns above.